# Identification of catalytic sites in cobalt-nitrogen-carbon materials for the oxygen reduction reaction

Andrea Zitolo [1], Nastaran Ranjbar-Sahraie[2], Tzonka Mineva[2], Jingkun Li [3], Qingying Jia [3], Serban Stamatin [4], George F. Harrington [5,6], Stephen Mathew Lyth [7,8], Petr Krtil [4], Sanjeev Mukerjee[3], Emiliano Fonda[1] & Frédéric Jaouen [2]

Single-atom catalysts with full utilization of metal centers can bridge the gap between molecular and solid-state catalysis. Metal-nitrogen-carbon materials prepared via pyrolysis are promising single-atom catalysts but often also comprise metallic particles. Here, we pyrolytically synthesize a Co–N–C material only comprising atomically dispersed cobalt ions and identify with X-ray absorption spectroscopy, magnetic susceptibility measurements and density functional theory the structure and electronic state of three porphyrinic moieties, $CoN_4C_{12}$, $CoN_3C_{10,porp}$ and $CoN_2C_5$. The $O_2$ electro-reduction and *operando* X-ray absorption response are measured in acidic medium on Co–N–C and compared to those of a Fe–N–C catalyst prepared similarly. We show that cobalt moieties are unmodified from 0.0 to 1.0 V versus a reversible hydrogen electrode, while Fe-based moieties experience structural and electronic-state changes. On the basis of density functional theory analysis and established relationships between redox potential and $O_2$-adsorption strength, we conclude that cobalt-based moieties bind $O_2$ too weakly for efficient $O_2$ reduction.

[1] Synchrotron SOLEIL, L'orme des Merisiers, BP 48 Saint Aubin, 91192 Gif-sur-Yvette, France. [2] Institut Charles Gerhardt Montpellier, UMR 5253, CNRS, Université Montpellier, Place Eugène Bataillon, 34095 Montpellier cedex 5, France. [3] Department of Chemistry and Chemical Biology, Northeastern University, Boston, MA 02115, USA. [4] J. Heyrovsky Institute of Physical Chemistry, Academy of Sciences of the Czech Republic, Prague, 18223, Czech Republic. [5] Center for Co-Evolutional Social Systems, Kyushu University, 744 Motooka, Nishi-ku, Fukuoka 819-0395, Japan. [6] Department of Materials Science and Engineering, Massachusetts Institute of Technology, 77 Massachusetts Avenue, Cambridge, MA 02139, USA. [7] International Institute for Carbon-Neutral Energy Research (WPI-I2CNER), Kyushu University, 744 Motooka, Nishi-ku, Fukuoka 819-0395, Japan. [8] Energy2050, Department of Mechanical Engineering, University of Sheffield, The Arts Tower, Sheffield S10 2TN, UK. Correspondence and requests for materials should be addressed to A.Z. (email: andrea.zitolo@synchrotron-soleil.fr) or to F.J. (email: frederic.jaouen@umontpellier.fr)

The transition from fossil to renewable energies is necessary to meet the rising energy demand while minimizing anthropogenic climate change and urban pollution[1–3]. Electrochemical energy conversion will play an increasing role for the storage of renewable electricity, production of fuels and their conversion into electricity. Hydrogen is an interesting energy vector since it can be produced via water electrolysis (hydrogen evolution reaction, HER) and later oxidized in $H_2$/air fuel cells to reform water and electricity on demand (hydrogen oxidation reaction, HOR). However, the electrochemical oxygen reactions (oxygen reduction reaction (ORR) and oxygen evolution reaction (OER)) are slow, limiting the roundtrip efficiency[4]. While acidic electrolytes are more restrictive than alkaline ones regarding the breadth of catalysts that may be stable under ORR/OER conditions, the advent of highly conductive and stable proton exchange membranes (PEM) has hitherto favoured the development of acidic fuel cells and electrolyzers. Their major drawback is, however, the need for platinum-group metals to catalyze the ORR and OER.

Following the pioneering report of Jasinski on the ORR activity of cobalt phthalocyanine[5], advanced Metal–N–C materials have since 1989 been prepared by pyrolyzing separate metal, nitrogen and carbon precursors[6]. Although the activity and durability of such catalysts have been improved[7–11], the identification of the active-site structure has lagged behind due to the non-crystallographic order of metal atoms in the most active sites and the simultaneous presence of crystalline metal phases. Metal ions coordinated with pyridinic nitrogen atoms embedded in a graphene matrix with hexagonal atomic arrangement, such as the $MeN_4C_{10}$ moiety, have for a long time been viewed as the most probable site structure[12–18]. Other studies hypothesized moieties integrated in disordered carbon sheets involving non-hexagonal rings, such as the $MeN_4C_{12}$ moiety[19, 20]. Calle-Vallejo et al.[21] investigated with density functional theory (DFT) the adsorption energy of oxygen intermediates on $MeN_4C_{12}$ and $MeN_4C_{10}$ moieties, revealing differences up to 0.7 eV. This highlights the importance of precisely determining the local site structures in Me–N–C materials for deciphering their reactivity.

Using X-ray absorption near-edge structure (XANES) spectroscopy we recently identified the active-site structure in pyrolyzed Fe–N–C catalysts as being a porphyrin-like $FeN_4C_{12}$ moiety, in contrast with $FeN_4C_{10}$ or $FeN_{2+2}C_{4+4}$ moieties previously assumed[22]. The formation of $FeN_4C_{12}$ moieties requires a strongly-disordered host material, in line with experimental observations[9, 23]. Although moieties present in Fe–N–C materials are now better described[22, 24–26], active sites in Co–N–C materials are still poorly identified. Co–N–C catalysts are ORR-active and advantageous vs. Fe–N–C since they produce less radical oxygen

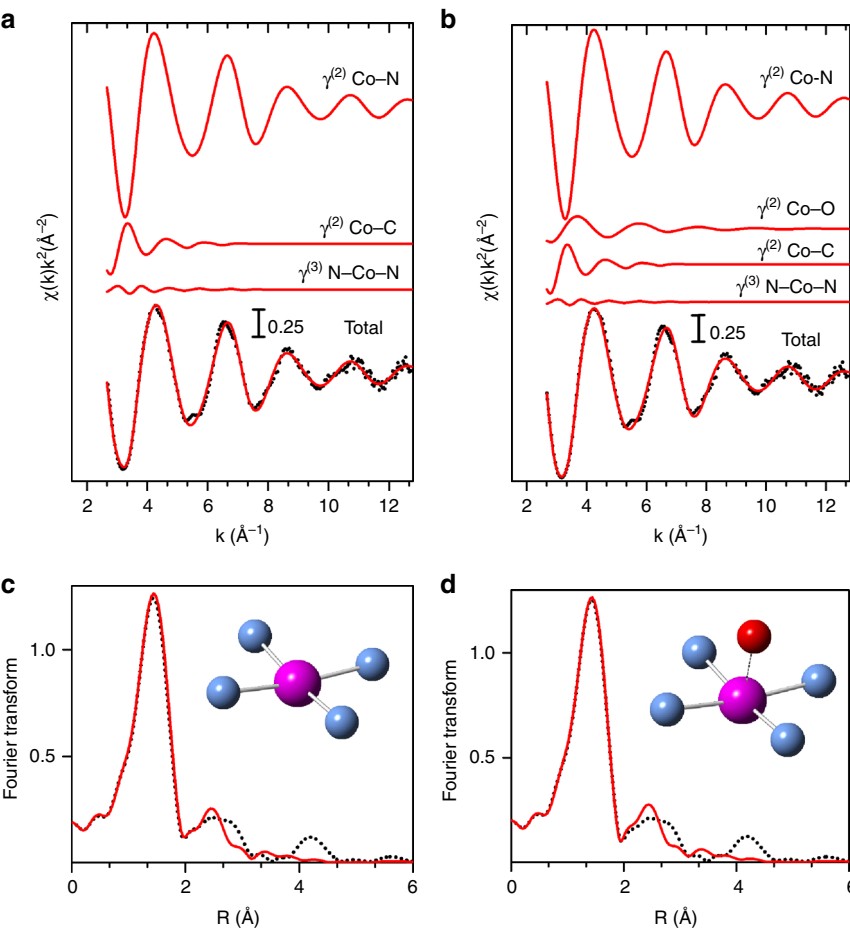

**Fig. 1** Co K-edge EXAFS analysis. Spectra of $Co_{0.5}$ with a $CoN_4$ moiety having zero (left) or one (right) oxygen atom in the axial direction. Cobalt, nitrogen and oxygen atoms are represented in purple, blue and red, respectively (carbon atoms in the second coordination sphere are not represented). **a, b** Curves from top to bottom: Co-N, Co-O and Co-C $\gamma^{(2)}$ two-body signals and the N-Co-N $\gamma^{(3)}$ three-body signal included in the fit, the total signal (red line) superimposed on the experimental one (black dots). **c, d** the fit in the Fourier-transformed space. No phase-shift correction was applied to the Fourier transforms

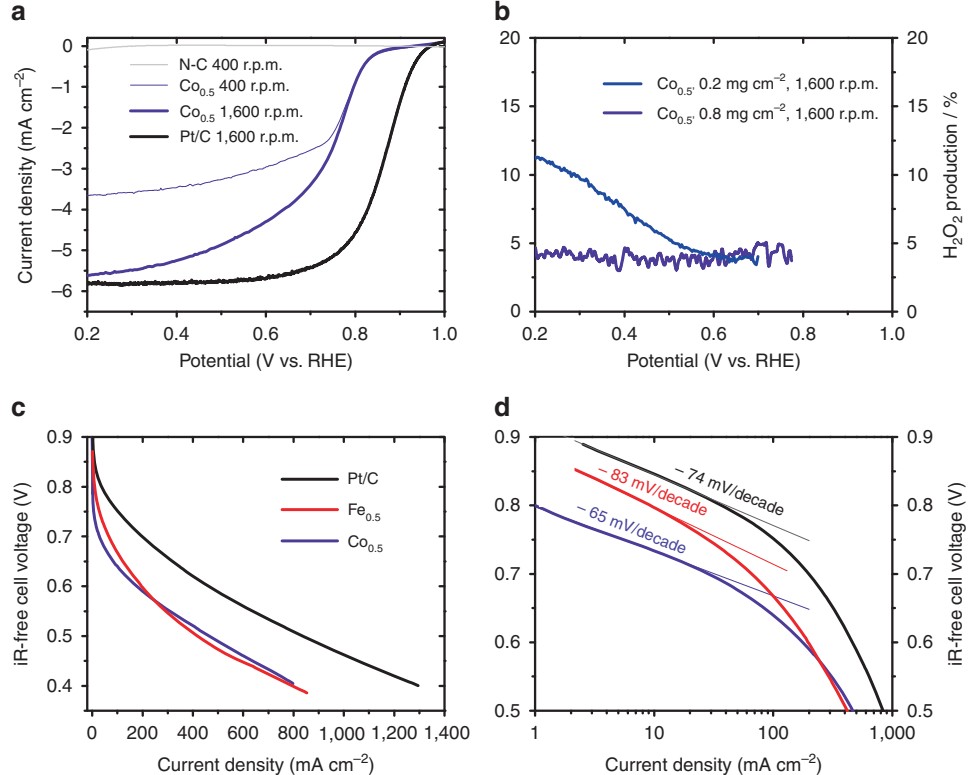

**Fig. 2** ORR electrochemical characterization. Measurements of $Co_{0.5}$, $Fe_{0.5}$, N-C and a Pt/C benchmark catalyst. Upper panels are RRDE measurements at various rotation rates and loadings performed in $O_2$-saturated 0.1 M $H_2SO_4$ aqueous solution. **a** ORR current density at the disk. The total catalyst loading for $Co_{0.5}$, N-C and Pt/C (5 wt% Pt/C) was 0.8 mg $cm^{-2}$. **b** % $H_2O_2$ measured during ORR at the ring. The total catalyst loading for $Co_{0.5}$ was 0.8 or 0.2 mg $cm^{-2}$. Lower panels are iR-corrected PEMFC polarization curves with $Co_{0.5}$, $Fe_{0.5}$ or Pt/C at the cathode, presented in linear scale **c** and semi-logarithmic scale **d**. The fuel cell temperature was 80 °C, pure $H_2$ and $O_2$ gases were fully humidified at the cell temperature, the gauge pressure was 1 bar. The cathode catalyst loading for $Co_{0.5}$ and $Fe_{0.5}$ was 4 mg $cm^{-2}$ (60 $\mu g_{metal}$ per $cm^2$). For Pt/C at the cathode, 5% Pt/C was diluted to 1% Pt with uncatalysed Vulcan carbon black, and 4 mg $cm^{-2}$ of 1% Pt/C (40 $\mu g_{Pt}$ per $cm^2$) deposited

species, leading to improved fuel cell durability[27]. They also have recently been reported to catalyze the HER[28, 29]. The existence of atomically dispersed cobalt atoms on N-doped graphene layers has been unambiguously demonstrated by mass spectroscopy and scanning transmission electron microscopy (STEM)[29, 30]. DFT investigations also concluded the energetically favoured formation of $CoN_xC_y$ moieties[20, 21, 31–34]. Extended X-ray absorption fine structure (EXAFS), previously used to investigate $CoN_xC_y$ moieties[35–37], is insufficiently sensitive to the spatial arrangement of coordinating light elements surrounding the absorbing cobalt nucleus.

The present study identifies with XANES the detailed structure of $CoN_xC_y$ moieties in a Co–N–C catalyst free of cobalt particles. Similar to Fe–N–C, the porphyrin-like $CoN_4C_{12}$ moiety is found to be the likeliest structure. In contrast to Fe–N–C, defective cobalt porphyrinic moieties are also viable candidates. Also metal-discriminating, *operando* XANES spectroscopy shows no change with electrochemical potential in the entire ORR range for Co–N–C, but a major change for Fe–N–C. The spectral change for Fe–N–C is correlated with a change of oxidation state in that potential region, while the lack of spectral change for cobalt is correlated with an unmodified oxidation state in that potential region.

## Results

### Absence of Co–Co bonds in $Co_{0.5}$ evidenced with EXAFS and TEM.

We synthesized a Co–N–C catalyst with the same approach developed for preparing Fe–N–C catalysts comprising only $FeN_xC_y$ moieties, and labeled it $Co_{0.5}$, with 0.5% Co in the catalyst precursor before pyrolysis (see Methods).[22] The EXAFS spectrum of $Co_{0.5}$ was analyzed assuming a variable number of nitrogen ligands in the equatorial plane and additional oxygen atoms in the axial position. The best-fit analysis is shown in Fig. 1, revealing 4- and/or 5-fold coordination structures. No Co–Co backscattering signal was needed to obtain an excellent fit. The dominant contribution to the EXAFS signal is given by the Co–N bonds of the first coordination shell (top of Fig. 1). The lower panels show the corresponding fits in the Fourier-transformed space. The peak at 1.45 Å is associated with the Co–N and/or Co–O first-shell contributions while the peak at 2.44 Å is due to Co–C. The structural parameters obtained from the fitting are reported in Supplementary Table 1. The Co–N distance, 1.95–1.96 Å, agrees with values determined by DFT for cobaltous-porphyrins.[38] However, the EXAFS experimental data can be reproduced with the same accuracy by a $CoN_4$ or an $O-CoN_4$ structure, as reported for $Fe_{0.5}$ (ref. [22]). Moreover, the uncertainty on the nitrogen coordination number (Supplementary Table 1) leaves room for hypothesizing moieties with a lower coordination, as investigated later in this study. The absence of metallic Co–Co bonds in $Co_{0.5}$ was independently confirmed by TEM (Supplementary Fig. 1). The selected area electron diffraction (SAED) pattern shows only broad rings, as expected for amorphous carbon. No sign of reflections originating from metallic particles were observed, and no evidence of particles was found in the images. Multiple areas of the sample were analyzed, and the

images and SAED pattern in Supplementary Fig. 1 are representative.

**Electrocatalytic properties of Co$_{0.5}$ towards the ORR.** The ORR activity and selectivity of Co$_{0.5}$ was then investigated with a rotating ring disk electrode (RRDE) in acid medium (Fig. 2a). The ORR current density at 0.8 V vs. a reversible hydrogen electrode (RHE) is 0.9 mA cm$^{-2}$ for a loading of 800 µg cm$^{-2}$ (12 µg$_{Co}$ cm$^{-2}$), which compares well with state of art Co–N–C catalysts in acid medium, in spite of low cobalt content in Co$_{0.5}$ (refs [39–41]). The Pt/C catalyst reaches 1.0 mA cm$^{-2}$ at 0.91 V vs. RHE, and its higher activity is due to its higher atom-specific ORR activity (the higher Pt loading, 40 µg$_{Pt}$ cm$^{-2}$, resulting in identical number of metal atoms per cm$^2$ as for cobalt, due to the 3.3 times higher molar mass of Pt vs. Co).

Regarding selectivity, the % peroxide released by Co$_{0.5}$ during ORR at 0.5–0.8 V vs. RHE is 3–5 % in 0.1 M H$_2$SO$_4$ at a loading of 800 µg cm$^{-2}$ (Fig. 2b). Co$_{0.5}$ is thus less selective to water formation than Fe$_{0.5}$, for which we measured <0.5 % H$_2$O$_2$ in acid, at equal catalyst loading.[22] The peroxide production was also measured at a lower loading, since it has been reported that, for some Fe–N–C catalysts, this can result in greatly increased detection of H$_2$O$_2$ (ref. [41]). The % H$_2$O$_2$ released by Co$_{0.5}$, however, only increased up to 11% at 200 µg cm$^{-2}$ (Fig. 2b). By analogy with what is known for unpyrolyzed Metal-N$_4$ macrocycles, this result may be explained by the generally higher formal potential of Metal$^{III}$/Metal$^{II}$ for cobalt vs. iron macrocycles, leading to weaker interaction with oxygen intermediates, including H$_2$O$_2$ (ref. [42]). Square-wave-voltammetry in acidic medium supports this possible explanation, identifying redox-peak positions at 1.25 V and 0.75 V vs. RHE for Co$_{0.5}$ and Fe$_{0.5}$, respectively (Supplementary Fig. 2). Although Co–N–C catalysts typically release more peroxide during the ORR, subsequent reactions between H$_2$O$_2$ and MeN$_x$C$_y$ moieties produce less radical oxygen species on cobalt- than iron-centers, which is beneficial for fuel cell durability[27]. Also, the insignificant ORR activity in acidic medium of N–C (Fig. 2a) means that the cobalt moieties are the most active-site for ORR in Co$_{0.5}$. This catalyst was then investigated in PEM fuel cell and compared to Fe$_{0.5}$ and Pt/C (Fig. 2c, d). For comparison, we prepared a Pt-based cathode with similar metal-site density (1% Pt/C) and similar thickness as Co$_{0.5}$ or Fe$_{0.5}$ cathodes. For carbon-rich catalysts, the thickness is set by the catalyst loading, here 4 mg cm$^{-2}$, typically resulting in 80–100-µm-thick layers[10]. Although the ORR activity at 0.8 V of Co$_{0.5}$ is lower than that of Fe$_{0.5}$ (Fig. 2d), the lower apparent Tafel slope (TS) observed with Co$_{0.5}$ at high potential leads to a higher power performance at E < 0.55 V. The apparent TS of 83 mV/decade for Fe$_{0.5}$ probably results from the outcome of an intrinsic TS with value > 83 mV per decade combined with the potential-dependence of the Fe(III) and Fe(II) coverages[43], with Fe(III) blocking the ORR due to overly strong O$_2$ adsorption. A similar phenomenon occurs for Pt, switching from surface-oxidized state to a surface free of oxygen adsorbates at ~ 0.8 V vs. RHE[44, 45]. In contrast, Co$_{0.5}$ does not change oxidation state throughout the ORR region (Supplementary Fig. 2) and the TS-value of 65 mV per decade observed at high potential is, therefore, intrinsic to the ORR mechanism on CoN$_x$C$_y$ moieties. Overall, and for similar cathode thickness, the gap between the curves for 1% Pt/C and Co$_{0.5}$ or Fe$_{0.5}$ is only a factor two at 0.6 V. A key issue is that, while Pt can be deposited as nanoparticles up to 50 wt% on carbon, the ORR activity of Metal–N–C materials levels off at 2–3 wt% metal[46]. Further work is thus needed to (i) improve reactant transport in thick Me–N–C electrodes, (ii) increase the number of active sites or (iii) their turnover frequency. The latter two tasks require identifying the structure of active moieties, object of the present study.

**Structural identification of Co-moieties in Co$_{0.5}$ by XANES.** To overcome the limitations of EXAFS, we resorted to XANES, which is more sensitive to the geometrical arrangement of atoms around the photo-absorber. Our XANES analysis was first validated with cobalt phthalocyanine (Co(II)Pc). The agreement between experimental and calculated spectra of Co(II)Pc is excellent in the whole energy range (Supplementary Fig. 3). For this fit, the square residual function, R$_{sq}$ (Methods section) is 1.06. The optimized structure resulted in a Co–N distance of 1.90(2) Å, which agrees with an X-ray diffraction determination[47]. We then calculated the Co K-edge XANES spectra of various CoN$_x$C$_y$ candidate sites for Co$_{0.5}$, including structures previously investigated with DFT[20, 31–34]. Some of these clusters are depicted in Supplementary Fig. 4 together with the results of the fitting procedure. Supplementary Fig. 4a, c report the XANES fitting performed assuming a CoN$_4$C$_{10}$ moiety enclosed in a graphene plane and a CoN$_{2+2}$C$_{4+4}$ moiety bridging two graphene planes, respectively. For the calculations, only the in-plane distances between cobalt and nitrogen atoms were allowed to vary. For these hypothetical sites, the fit between experimental and calculated XANES spectra is unsatisfactory (R$_{sq}$ = 2.97 and 2.81, respectively) and the fit quality was little improved by adding an axial oxygen (Supplementary Fig. 4b, d). The disordered arrangement of nitrogen and carbon atoms surrounding cobalt ions has been recently revealed by STEM for another Co–N–C catalyst[29]. This suggests that the arrangement of carbon atoms around cobalt ions is different from that in graphene.

We then considered two defective sites derived from CoN$_4$C$_{10}$, namely pyridinic CoN$_3$C$_{10,pyr}$ and CoN$_2$C$_8$ (Supplementary Fig. 5a, c) and two defective sites derived from CoN$_{2+2}$C$_{4+4}$, namely CoN$_{2+1}$C$_{4+3}$ and CoN$_2$C$_4$ (Supplementary Fig. 5e, g). None of these four pyridinic-defective candidate sites correctly reproduced the experimental XANES spectrum of Co$_{0.5}$. Again, the addition of an O$_2$ molecule in end-on mode did not improve much the fit quality (Supplementary Fig. 5b, d, f, h). The structural parameters obtained from these analyses are summarized in Supplementary Table 2.

We then considered a cobalt moiety based on a porphyrinic architecture, analogous to that found for the Fe-based active-site geometry in Fe$_{0.5}$ (ref. [22]). Fig. 3a, b show the XANES analyses performed on the square-planar CoN$_4$C$_{12}$ moiety, with and without an axial O$_2$ adsorbed end-on. These fits show an excellent agreement between experimental and theoretical spectra and correspond to a moiety where the cobalt ion is coordinated in-plane by four nitrogen atoms at 1.96–1.97 Å, and possibly by one oxygen molecule at 2.23 Å (Table 1, rows 1 and 2). Two defective sites, derived from the porphyrinic CoN$_4$C$_{12}$ moiety by subtracting one or two nitrogen atoms (porphyrinic CoN$_3$C$_{10,porp}$ and CoN$_2$C$_5$ model sites), also correctly reproduce the experimental spectrum (Fig. 3c, e, respectively). For the CoN$_3$C$_{10,porp}$ motif, the XANES analysis resulted in a Co–N bond length of 1.96 Å when O$_2$-free, and 1.99 Å when O$_2$ is adsorbed end-on without out-of-plane displacement of cobalt (Table 1, rows 3 and 4). The XANES analysis also revealed the CoN$_2$C$_5$ edge-defect, representing a halved porphyrinic moiety, binding O$_2$ end-on with a Co–O bond length of 1.90 Å with O$_2$ in the same plane as the CoN$_2$C$_5$ motif (Fig. 3e, f and Table 1, rows 5 and 6). For the CoN$_2$C$_5$ motif only, the bound O$_2$ molecule was necessary to reach a good fit (R$_{sq}$-value of 1.12–1.16, Table 1). For the CoN$_4$C$_{12}$ and CoN$_3$C$_{10,porp}$ moieties (with or without adsorbed O$_2$), the Co–N bond lengths determined by XANES and EXAFS are in good agreement, and so

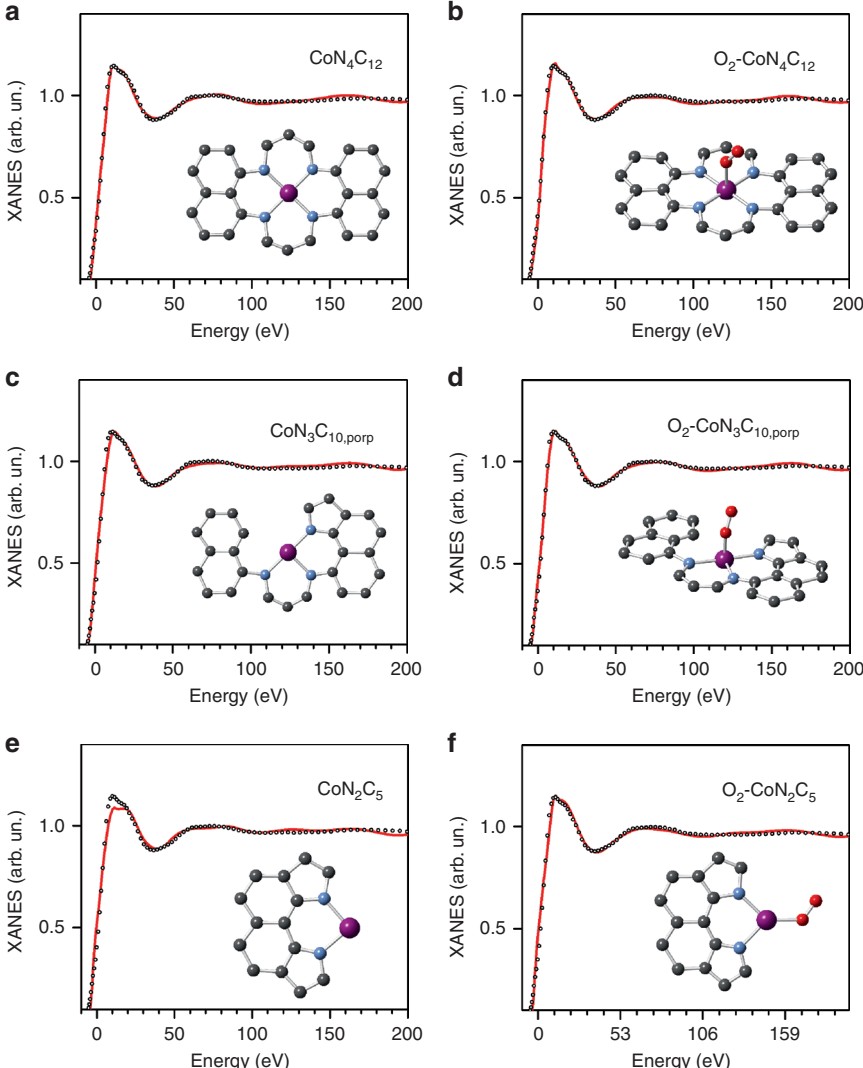

**Fig. 3** Experiment versus theory. **a–f** Comparison between the K-edge XANES experimental spectrum of $Co_{0.5}$ (black hollow circles) and the theoretical spectrum calculated with the depicted structures (solid red lines). Cobalt, nitrogen, oxygen and carbon atoms are represented in purple, blue, red and gray, respectively

is the case for the $CoN_2C_5$ moiety free of adsorbed $O_2$ (Table 1 and Supplementary Table 1).

The energy of formation and $O_2$ adsorption energy were then calculated by DFT-D approach for the possible sites identified by XANES analysis, namely $CoN_4C_{12}$, $CoN_3C_{10,por}$ and $CoN_2C_5$, and the results were compared to those obtained for the $CoN_4C_{10}$ pyridinic moiety (Supplementary Fig. 6). The optimized bond distances for the above-mentioned porphyrinic sites, with and without a dioxygen molecule adsorbed in end-on mode, follow the trends of bond distances determined by EXAFS and XANES (Supplementary Table 3). The stability of these porphyrinic defective motifs is assessed by the Co(II) binding energy, −6.8 to −7.5 eV for the ground states, comparable to −8.2 eV obtained for $CoN_4C_{10}$ (Supplementary Table 4). Moreover, our DFT-D results reveal that the adsorption of an $O_2$ molecule results in the displacement of the cobalt ion by 0.4 and 0.5 Å above the plane for $CoN_3C_{10,porp}$ and $CoN_2C_5$, respectively. The $O_2$ adsorption energies in the ground-state structures are −0.80, −1.23 and −1.26 eV for $CoN_4C_{12}$, $CoN_3C_{10,porp}$ and $CoN_2C_5$, respectively. This is comparable to the $O_2$ adsorption energies for the ground state of the $CoN_4C_{10}$ site, −0.97 eV. Although $O_2$ adsorbs on all

cobalt moieties, the obtained adsorption energies are, however, significantly smaller than those of the $FeN_4C_{10}$ and $FeN_4C_{12}$ moieties in ground state, −1.83 and −1.84 eV, respectively (Supporting Information of ref. [22]). This indicates that cobalt binds $O_2$ more weakly compared to iron-based moieties. Due to the lower experimental ORR activity of $Co_{0.5}$ vs. $Fe_{0.5}$ (Fig. 2d) one can unambiguously conclude that cobalt moieties are situated on the weak-binding side of a volcano plot. While no definitive conclusion can be made on the volcano-plot position of iron moieties in $Fe_{0.5}$ only from the present work (the weak and strong binding branches being both possible), yet unpublished work on $Fe_{0.5}$ post-treated with $H_2O_2$ suggests that $Fe_{0.5}$ is also positioned on the weak binding side of a volcano plot. The weaker binding of $O_2$ (and thus of $H_2O_2$, via scaling relationships) by $CoN_xC_y$ moieties also explains the higher amount of peroxide detected by RRDE for Co–N–C vs. Fe–N–C, as reported above. The $O_2$ adsorption energy of molecular cobalt catalysts is often weaker than desired for an ideal metal-centered moiety,[42] and the same conclusion is made from the present combined experimental and theoretical study for the pyrolyzed $Co_{0.5}$ catalyst. Hence, cobalt-moieties with the highest calculated $O_2$ adsorption energy should

**Table 1 XANES structural parameters for porphyrinic cobalt moieties**

| Row nbr. | Moiety | Co–N/Å | Co–O/Å | Bending/° | $R_{sq}$ |
|---|---|---|---|---|---|
| | $CoN_4C_{12}$ | | | | |
| 1 | 4-fold | 1.96 (2) | — | — | 1.12 |
| 2 | 5-fold | 1.97 (3) | 2.23 (5) | 33 (6) | 1.08 |
| | $CoN_3C_{10,porp}$ | | | | |
| 3 | 3-fold | 1.96 (3) | — | — | 1.16 |
| 4 | 4-fold | 1.99 (3) | 1.86 (4) | 32 (7) | 1.05 |
| | $CoN_2C_5$ | | | | |
| 5 | 2-fold | 1.96 (3) | — | — | 1.98 |
| 6 | 3-fold | 2.01 (4) | 1.90 (4) | 50 (5) | 1.13 |

Abbreviation: XANES, X-ray absorption near-edge structure
Best-fit structural parameters obtained from the analysis of the XANES spectrum of $Co_{0.5}$ performed on the structures proposed in this work and depicted in Fig. 3. Bending is the angle between the Co–O vector and the O–O bond and $R_{sq}$ is the residual function. Errors are given in parentheses

be the most active ones, which gives the following expected ORR turnover frequency ranking $CoN_2C_{5} > CoN_3C_{10,porp} > CoN_4C_{12}$. To better assess which moieties may be present in $Co_{0.5}$, we measured the magnetic susceptibility of $Co_{0.5}$ and N–C with superconducting quantum interference device (Supplementary Fig. 7). The plot of the inverse of the magnetic molar susceptibility $\chi_m$ of $Co_{0.5}$ is linear with temperature (inset of Supplementary Fig. 7), as expected for Curie–Weiss paramagnetism. The positive intercept at 0 K leads to a negative Curie–Weiss temperature of −0.36 K, signifying weak antiferromagnetic interaction between magnetic moments[48]. The value of the slope $1/\chi_m = f(T)$ yields a magnetic moment $\mu_{eff} = 3.52$ Bohr magnetons $(\mu_B)$, averaged on all cobalt moieties in $Co_{0.5}$. For the first-row transition metals, to the first approximation, only the spin contribution to the effective magnetic moment can be considered $(\mu_{eff} = 2 \mu_B \sqrt{(s^2 + s)})$ leading to $s_{average} = 1.33$ in $Co_{0.5}$, where $s_{average}$ is the spin density averaged on all cobalt moieties present in $Co_{0.5}$. According to the DFT spin density analysis and selecting the three best candidates on the basis of EXAFS and XANES analysis (Supplementary Table 4, $CoN_4C_{12}$, $CoN_3C_{10,porp}$ and $O_2$-$CoN_2C_5$ in their ground state, spin density of cobalt of 0.88, 0.58 and 1.83, respectively), one can propose that $Co_{0.5}$ contains for example $CoN_4C_{12}$ and $O_2$-$CoN_2C_5$ in a 53%/47% distribution, or $CoN_3C_{10,porp}$ and $O_2$-$CoN_2C_5$ in a 40%/60% distribution. Both distributions result in a theoretical $s_{average}$ value of 1.33, similar to the experimental value (Supplementary Table 5). Considering only these three motifs, the fraction of $O_2$-$CoN_2C_5$ sites must be in the range of 47–60%, the remainder being split between $CoN_4C_{12}$ and $CoN_3C_{10,porp}$ sites.

We recently demonstrated that the porphyrinic $FeN_4C_{12}$ moiety with one or two axial ligands matched the XANES experimental spectrum of $Fe_{0.5}$ and other Fe-based catalysts[22]. Since defective Fe-porphyrinic moieties were not investigated in that study, we extended here the XANES analysis of $Fe_{0.5}$ in order to investigate whether such Fe-based moieties could also match the experimental XANES spectrum of $Fe_{0.5}$. The analyses are shown in Supplementary Fig. 8 for $FeN_3C_{10,porp}$ and $FeN_2C_5$. While $O_2$-$FeN_3C_{10,porp}$ reproduces the general features of the spectrum, the fit quality is poorer than that obtained with $O_2$-$FeN_4C_{12}$ (Fig. 5e, f in ref. [22]). Although this does not preclude the existence of $FeN_3C_{10,porp}$ moieties in $Fe_{0.5}$, they may only represent a minor iron fraction. The present study therefore suggests that the structural disparity of $MeN_xC_y$ moieties is larger in Co–N–C than Fe–N–C materials. Support for this is also provided by square-wave voltammetry (SWV) revealing a broader peak for $Co_{0.5}$ (Supplementary Fig. 2).

**Operando XANES signatures of Co- and Fe-moieties during ORR.** We then investigated the behavior of Co- and Fe-based moieties in *operando* conditions. Figure 4a shows the normalized *operando* XANES spectra recorded at ORR potential range for $Co_{0.5}$ in acid medium and Fig. 4b shows the same type of data for $Fe_{0.5}$. Although the latter show a large change with electrochemical potential, similar to a previous report by Mukerjee's group on other Fe catalysts[43], the magnitude of the change is insignificant for $Co_{0.5}$ in the ORR region, even on the differential $\Delta\mu$ spectra (inset of Fig. 4a). The potential dependence of the XANES spectra observed on $Fe_{0.5}$ in $N_2$-saturated electrolyte supports the fact that the XANES variability is primarily controlled by the electrochemical potential, while the adsorption of $O_2$ or oxygen intermediates has only a minor effect (Fig. 4d). The spectral changes observed in Fig. 4b probably result not only from a change in oxidation state, which would a priori only shift the edge position of the XANES spectra, but also from deeper structural changes and reorganization of the N (or C) ligands, as proposed in ref. [26] and/or an Fe(II) low-to-high spin-crossover[49]. SWV (Supplementary Fig. 2) shows that, while $Fe_{0.5}$ displays a clear peak at 0.75 V vs. RHE in acidic medium, $Co_{0.5}$ did not show any signal in the 0–1 V region, but instead shows a redox transition at 1.25 V vs. RHE. This largely explains the different potential-dependences of the XANES spectra recorded in the ORR region for $Fe_{0.5}$ and $Co_{0.5}$. In addition, the redox potential observed for $Fe_{0.5}$ corresponds to its onset potential for the ORR (see Fig. 3b in ref. [22]). It is however surprising that the XANES spectra of $Fe_{0.5}$ are still changing below 0.6 V vs. RHE, while most Fe ions should already be in +II oxidation state. Spin-state or conformation changes may still be occurring below 0.6 V.

The very different position of the redox peak for $Fe_{0.5}$ and $Co_{0.5}$ experimentally supports the fact that $O_2$ adsorption on cobalt moieties is much weaker than on iron moieties. Linear relationships between the Me(III)/Me(II) redox potential and the adsorption energy for $O_2$ have clearly been established for various macrocycles comprising first-row transition metals (Fig. 3 in ref. [42]). On this basis, the experimentally-determined relative positions of the redox peak for $Fe_{0.5}$ and $Co_{0.5}$ (0.75 and 1.25 V vs. RHE) are in line with our DFT calculations that predict $O_2$ adsorption energies of ~ −1.8 eV for $FeN_4C_{12}$ and −0.8 to −1.2 eV for the porphyrinic Co-moieties.

Comparative *operando* XANES spectra collected under $O_2$ and $N_2$ provide additional insights in the reactivity of the active sites. Fig. 4c, d show the XANES spectra measured in $O_2$- and $N_2$-saturated acid electrolyte at two representative potentials (0.2 and 0.8 V vs. RHE) for $Co_{0.5}$ and $Fe_{0.5}$, respectively. Although for $Fe_{0.5}$ we do not observe significant spectral changes in $O_2$-free or $O_2$-saturated electrolyte, for $Co_{0.5}$ a clear variation is observed between 7720 and 7735 eV. This effect is magnified in the experimental $\Delta\mu_{O2-N2}$ spectra obtained by subtracting from the XANES spectrum of $Co_{0.5}$ measured under $O_2$ that measured under $N_2$ (insets of Fig. 4c, d). A fit of the $\Delta\mu_{O2-N2}$ spectrum of $Co_{0.5}$ at 0.8 V is shown in Supplementary Fig. 9, corresponding to an active-site structure with four nitrogen atoms at 1.95 Å with or without an oxygen molecule adsorbed end-on at 2.22 Å. In $N_2$-saturated electrolyte, oxygen comes exclusively from water activation, whereas in $O_2$-saturated electrolyte the oxygen may come from water activation or $O_2$ adsorption. These $\Delta\mu$ analyses strongly suggest that, for $Fe_{0.5}$, an Fe-O bond probably exists in both $O_2$-saturated and $N_2$-saturated electrolytes, with oxygen originating from $O_2$ and $H_2O$, respectively. In contrast, for $Co_{0.5}$ the active sites are less oxophilic and a Co–O bond is formed only in $O_2$-saturated electrolyte. To the best of our knowledge, this is the first direct *operando* evidence of molecular $O_2$ adsorption on the cobalt centers in Co–N–C materials. The general picture that emerges from the present work supports the

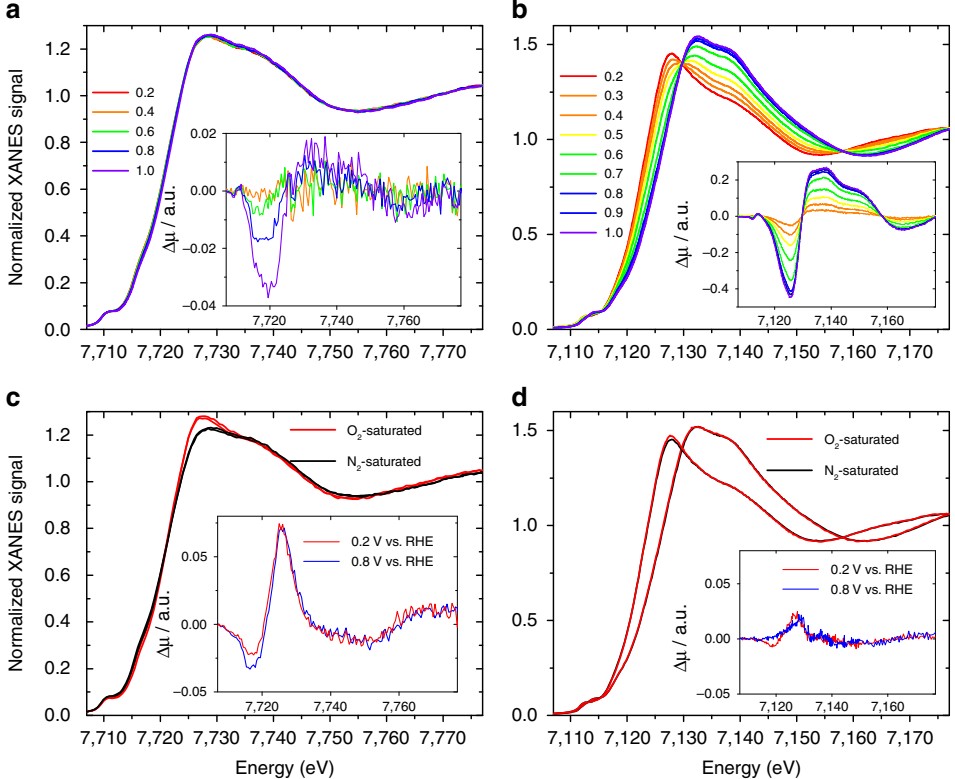

**Fig. 4** *Operando* XANES spectra. Taken in 0.5 M $H_2SO_4$ for $Co_{0.5}$ (left) and $Fe_{0.5}$ (right). The spectra were measured as a function of the electrochemical potential in $N_2$-saturated electrolyte for **a** $Co_{0.5}$ and **b** $Fe_{0.5}$ (legend for potentials is in V vs RHE) and measured as a function of the saturating gas ($O_2$ or $N_2$) at either 0.2 V vs. RHE or 0.8 V vs. RHE for **c** $Co_{0.5}$ and **d** $Fe_{0.5}$. Insets of **a** and **b** are differential $\Delta\mu$ XANES spectra obtained by subtracting the normalized spectrum at every potential to the spectrum recorded at 0.2 V vs. RHE. Insets of **c** and **d** are $\Delta\mu$ spectra obtained by subtracting the normalized XANES spectra recorded in $O_2$- to that recorded in $N_2$-saturated electrolyte, at a fixed potential

view that the active moieties formed at high temperature have a porphyrinic or defective porphyrinic architecture and also obey to the same general trends that have been reported on well-defined macrocycles comprising 3d transition-metal ions from the first row[22, 26, 42, 43]. However, while cobalt moieties in pyrolyzed materials do not change structure when catalyzing the ORR in acid medium, the iron moieties do change structure due to the change of oxidation and/or the spin state. The exact structural change induced by the redox switch will require extensive modeling work, which is beyond the scope of the present paper. Last, while general reactivity trends seem to apply for both pyrolyzed Metal–N–C materials and well-defined metal macro-cycles, this does not mean however that their absolute ORR activity and $O_2$ adsorption energy are the same. For example, the $O_2$ adsorption energy on the three porphyrinic-derived $CoN_xC_y$ moieties in ground state is –0.80 to –1.26 eV (Supplementary Table 4), higher than that reported for cobalt phthalocyanines and porphyrins (–0.60 to –0.35 eV).[42] The covalent integration of the Metal-$N_x$ moieties in a conductive carbon matrix generally modifies the electron density at the metal relative to a metal macrocycle adsorbed on carbon.[50]

**Electroactivity of cobalt moieties for other reactions**. After having identified the porphyrinic structure of three possible cobalt moieties in $Co_{0.5}$, we investigated their electrochemical activity for the OER, HER and HOR in acid medium. The cobalt moieties are the most active-site for OER in $Co_{0.5}$, as indicated by the significantly lower OER activity of N–C (purple vs. Gray curve in Supplementary Fig. 10). Due to the low density of cobalt

sites in $Co_{0.5}$, the OER potential at a current density of 2 mA cm$^{-2}$ is however positively shifted by 200 mV relative to that of unsupported $IrO_2$ nanoparticles. While the OER activity of such cobalt moieties is interesting from a fundamental point of view, the low stability of carbon at such high electrochemical potential limits their practical application.[51] The low current observed at 1.3–1.6 V vs. RHE on $Co_{0.5}$ might in fact entirely or partially originate from carbon corrosion, enhanced in the presence of cobalt relative to N-C. OER activity has been predicted from DFT theory for $CoN_xC_y$ and $FeN_xC_y$ moieties.[52] In contrast to the case for ORR, the oxygen-binding energy of the porphyrinic $CoN_4C_{12}$ moiety is close to the ideal value that may be expected for a Metal-$N_4$ moiety.[52] OER activity has also been reported experi-mentally for heterogeneous Co–N[52]C materials in alkaline med-ium.[22] These heterogeneous catalysts mostly comprised metallic cobalt or cobalt oxide while the presence of $CoN_xC_y$ moieties could not be certified.[53–55] The onset of OER in acidic medium on $Co_{0.5}$ is not well defined but clearly situated above the redox peak associated with Co(II)/Co(III) at 1.25 V vs. RHE (Supple-mentary Fig. 2), suggesting Co(III)$N_xC_y$ moieties could be the active-site for OER in $Co_{0.5}$. The *operando* XANES spectra were also recorded at OER potentials, and these spectra showed only a small difference compared to the XANES spectrum recorded at 1.0 V vs. RHE (Supplementary Fig. 11), supporting the fact that $CoN_xC_y$ moieties were still present at such high potential and are responsible for the OER activity of $Co_{0.5}$. As for the ORR, it must be noted that the cobalt content in $Co_{0.5}$ was not optimized to reach the highest activity.

In contrast, Co–N–C materials should be stable for long duration at low potentials needed to catalyze the HER. Two

recent reports demonstrated the high activity of $CoN_xC_y$ moieties toward HER in acid and alkaline medium.[28, 29] The HER polarization curve of $Co_{0.5}$ is however negatively shifted by 200 mV relative to Pt/C (Supplementary Fig. 12). In addition, $Co_{0.5}$ did not show any activity for the HOR (Supplementary Fig. 13), highlighting an irreversible behavior between HER and HOR on cobalt moieties. This intriguing fact seems to also apply to non-precious metal solid-state materials such as Mo and W nitrides, sulfides, and carbides. Such structures lead to efficient HER catalysts but were never reported to be efficient HOR catalysts.[56, 57]

## Discussion

We identified with XANES, EXAFS, magnetic susceptibility measurements and DFT analysis the detailed structures and electronic state of single-atom catalytic sites in pyrolyzed Co–N–C materials. Three (defective) porphyrinic moieties were identified and showed some activity toward the ORR, OER and HER in acidic medium. DFT-D calculations showed that these cobalt moieties likely bind oxygen intermediates too weakly relative to an optimum Metal–N–C ORR catalyst, while iron moieties bind oxygen intermediates more strongly than cobalt ones. It however cannot be determined from the present study if iron moieties bind oxygen intermediates more or less strongly than optimally desired for the ORR. These insights support that the general activity descriptors for the active sites in pyrolyzed Metal–N–C catalysts are similar to those identified earlier for molecular Metal-N$_4$ macrocycles. For Co–N–C catalysts, modifications of the carbon matrix with nitrogen or other light elements resulting in a stronger $O_2$ binding energy at cobalt centers should improve their ORR activity. Generally, rational modification of the carbon matrix hosting such metal-centered moieties could be used for optimizing their activity toward a large variety of (electro)chemical reactions. These materials can bridge the gap between molecular and solid-state catalytic materials, offering perspectives for catalysts based on earth-abundant elements.

## Methods

**Synthesis**. The catalyst precursor of $Co_{0.5}$ was prepared from a Zn(II) zeolitic imidazolate framework (Basolite Z1200 from BASF, labeled ZIF-8), Co(II) acetate and 1,10-phenanthroline. Weighed amounts of the powders of Co(II)Ac (15.97 mg), phen (200 mg) and ZIF-8 (800 mg) were ball-milled (Pulverisette 7 premium, Fritsch) at 400 rpm for 2 h (4 × 30 min with 5 min rest in-between) in a $ZrO_2$ crucible (45 cm$^3$) filled with 100 $ZrO_2$ balls (5 mm diameter). A split-hinge oven was equilibrated at 1,050 °C under Ar flow, and the catalyst precursor was quickly introduced and pyrolyzed at 1,050 °C in Ar for 1 h (see details in ref. [22]). The N–C material was prepared similarly but without cobalt. Due to a mass loss of 65–70 wt % during pyrolysis in Ar (unmodified by the presence of Co or Fe at 0.5 wt%) caused by volatile products formed from ZIF-8 and phen while Co does not form volatile compounds, the cobalt content in $Co_{0.5}$ is 1.5 wt%.

**Rotating ring disk electrode measurements**. Electrochemical activity and selectivity toward the ORR was determined using the RRDE technique. For $Co_{0.5}$, N–C and Pt/C (5 wt% Pt/C, provided by Johnson Matthey) catalysts, an ink including 10 mg of catalyst, 109 μl of a 5C wt% Nafion® solution containing 15–20% water, 300 μl of ethanol and 36 μl of de-ionized water was sonicated and mixed using a vortex. An aliquot of 9 μl was deposited on the glassy-carbon disk (0.247 cm$^2$) with a Pt ring (Pine Instruments, nominal collection efficiency 0.37), resulting in a loading of 818 μg·cm$^{-2}$. For lower $Co_{0.5}$ loadings (200 μg cm$^{-2}$), the catalyst mass and Nafion solution aliquot in the ink were proportionally decreased, and the decrease in Nafion solution balanced by increased ethanol aliquot. 9 μl of ink was again deposited on glassy carbon. The working electrode with the deposited catalyst layer was used in a four-electrode cell setup connected to a bipotentiostat (Biologic SP 300) and rotator (MSR, Pine Instruments). The counter and reference electrodes were a graphite rod and a reversible hydrogen electrode (RHE), respectively. The acidic electrolyte was an $O_2$-saturated aqueous solution of 0.1 M $H_2SO_4$, except for ORR on Pt/C (0.1 M $HClO_4$). For Pt/C only, the electrode was first cleaned by performing 300 cycles at 500 mV s$^{-1}$ between 0 and 1 V vs. RHE. The RDE polarization curves were recorded with a scan rate of 10 (Fe–N–C) or 50 mV s$^{-1}$ (Pt/C, to avoid contamination) at 1,600 r.p.m. and corrected for the

background current measured in $N_2$-saturated electrolyte. The second cycle was used for correction. For measuring the % $H_2O_2$ released during ORR, the potential of the Pt ring was held at 1.2 V vs. RHE.

**Square-wave voltammetry**. Catalysts inks were prepared by dispersing 3.1 mg catalyst in a solution prepared by mixing 150.2 μL of millipore water, 465 μL of isopropyl alcohol and 6.2 μl of 5 wt% Nafion® (Nafion to catalyst mass ratio of 10 wt%). The ink solution was then sonicated 60 min. 20 μl of the ink was pipetted on the glassy carbon disk (0.247 cm$^2$) to reach a loading of 400 μg cm$^{-2}$. This lower loading than used for most RRDE experiments was necessary to obtain a proper balance between the signal coming from the redox peak and that due to the double layer. The electrodes were dried through a rotational drying method at 800 r.p.m. SWV experiments were carried out in 0.1 M $HClO_4$ aqueous electrolyte using an Autolab bipotentiostat (PGSTAT302N) with a step potential of 5 mV, potential amplitude of 20 mV and scan frequency of 10 Hz in a standard electrochemical cell (Chemglass).

**Fuel cell measurements**. For the membrane electrode assembly, cathode inks were prepared using the following formulation: 20 mg of catalyst, 652 μl of a 5 wt.% Nafion® solution containing 15–20% water, 326 μl of ethanol and 272 μl of de-ionized water. The inks were alternatively sonicated and agitated with a vortex mixer every 15 min, for a total of 1 h. Then, three aliquots of 405 μl of the catalyst ink were successively deposited on the microporous layer of an uncatalysed 4.84 cm$^2$ gas diffusion layer (Sigracet S10-BC) to reach a catalyst loading of 4 mg cm$^{-2}$. The cathode was then placed in a vacuum oven at 80 °C to dry for 2 h. The anode used for all PEM fuel cell tests performed in this work was 0.5 mg$_{Pt}$·cm$^{-2}$ on Sigracet S10-BC. Assemblies were prepared by hot-pressing 4.84 cm$^2$ anode and cathode against either side of a Nafion NRE-211 membrane at 135 °C for 2 min. PEMFC tests were performed with a single-cell fuel cell with serpentine flow field (Fuel Cell Technologies Inc.) using an in-house fuel cell bench and a Biologic Potentiostat with 50 A load and EC-Lab software. For the tests, the fuel cell temperature was 80 °C, the humidifiers were set at 85 °C, and the inlet pressures were set to 1 bar gauge for both anode and cathode sides. The flow rates for humidified $H_2$ and $O_2$ were controlled downstream of the polymer electrolyte membrane fuel cell (PEMFC) and fixed at 50 s.c.c.m. Polarization curves were recorded by scanning the cell voltage at 0.5 mV s$^{-1}$.

**Spectroscopic experimental characterization**. Co and Fe K-edge X-ray absorption spectra were collected at room temperature at SAMBA beamline (Synchrotron SOLEIL) equipped with a double crystal Si 220 monochromator. Catalyst inks were prepared by mixing 10 mg catalyst with 50 μl de-ionized water and 100 μL of 5 wt% Nafion® solution with ultrasounds. A 50 μl aliquot was then pipetted on ~3 cm$^2$ circular area of a 100-μm-thick graphite foil (Goodfellow cat. C 000200/2), resulting in a catalyst loading of ~1 mg cm$^{-2}$. The graphite foil then served as a working electrode, and was installed in an electrochemical cell (PECC2, from Zahner), see Supplementary Fig. 14. The cell also includes a reference Ag/AgCl electrode, and a Pt counter electrode. The cell was filled with a given electrolyte and saturated with either $O_2$ or $N_2$ by continuously bubbling gas in the electrolyte. *Operando* measurements were performed by recording the $K_\alpha$ X-ray fluorescence of the element under study (Co or Fe) with a Canberra 35-elements monolithic planar Ge pixel array detector, while the ex situ spectra of the reference compound Co(II)Pc and $Co_{0.5}$ were recorded in transmission geometry on pelletized disks of 10 mm diameter with 1 mm thickness, using Teflon powder (1 μm particle size) as a binder.

**Spectroscopic analysis and modeling**. The EXAFS data analysis was performed with the GNXAS code[58, 59]. In that approach the interpretation of the experimental data is based on the decomposition of the EXAFS $\chi(k)$ signal (defined as the oscillation with respect to the atomic background cross-section normalized to the corresponding K-edge channel cross-section) into a summation over n-body distribution functions $\gamma^{(n)}$ calculated by means of the multiple-scattering (MS) theory. The cobalt first coordination shells have been modeled with Γ-like distribution functions which depend on four parameters, namely, the coordination number N, the average distance R, the mean-square variation $\sigma^2$ and the skewness $\beta$. Note that $\beta$ is related to the third cumulant $C_3$ through the relation $C_3 = \sigma^3\beta$. The standard deviations given for the refined parameters in Supplementary Table 1 are obtained from $k^2$-weighted least-squares refinements of the EXAFS function $\chi(k)$, and do not include systematic errors of the measurements. Least-square fits of the EXAFS raw experimental data have been performed by minimizing a residual function of the type:

$$R_i(\{\lambda\}) = \sum_{i=1}^{N} \frac{\left[\alpha_{exp(E_i)} - \alpha_{mod}\left(E_i; \lambda_1, \lambda_2, \ldots, \lambda_p\right)\right]^2}{\sigma_i^2}, \tag{1}$$

where $N$ is the number of experimental points, $E_i$ $\{\lambda\} = (\lambda_1, \lambda_2,\ldots, \lambda_p)$ are the $p$ parameters to be refined and $\sigma_i^2$ is the variance associated with each experimental point $\alpha_{exp}(E_i)$. Additional non-structural parameters were minimized, namely $E_0$

(core ionization threshold energy) and $S_0^2$ (amplitude reduction factor taking into account intrinsic losses).

The XANES data analysis of $Co_{0.5}$ was carried out with the MXAN code in the framework of the full MS scheme, following the same modeling approach applied to identify the structure of the catalytic active sites in Fe–N–C materials[22]. The MXAN method is based on the MT approximation for the shape of the potential and uses a complex optical potential, based on the local density approximation of the self-energy of the excited photoelectron[60]. The minimization of the Co(II)Pc XANES spectrum has been carried out starting from the X-ray structure of this compound[47], while in the case of $Co_{0.5}$ we tested different coordination geometries. The cluster size used in the calculations was chosen on the basis of a convergence criterion. The fit includes a minimal number of selected parameters: the Co–N and Co–O distances, the Co-displacement, the Co-5th ligand distance and the bending angle between the Co–O vector and the O–O bond. During the fit, the outer carbon atoms of the ring rigidly followed the motion of the nitrogen atoms. Least-square fits of the experimental data in the space of the structural parameters were achieved by minimizing the residual function defined as

$$R_{sq} = \frac{n}{\frac{m}{\sum\limits_{i=1}^{m} w_i}} \sum\limits_{i=1}^{m} \frac{w_i(y_i^{th} - y_i^{exp})^2}{\varepsilon_i^2}, \qquad (2)$$

where $n$ is the number of independent parameters, $m$ is the number of data points, $y_i^{th}$ and $y_i^{exp}$ are the theoretical and experimental values of absorption, respectively, $\varepsilon_i$ is the individual error in the experimental data set, and $w_i$ is a statistical weight. Here, we assumed a constant experimental error, $\varepsilon = 1.2\%$, for the whole experimental data set. Five non-structural parameters have been optimized, namely the Fermi energy level $E_F$, the experimental resolution $\Gamma_{exp}$, the threshold energy $E_0$ and energy and amplitude of the plasmon, $E_s$ and $A_s$. The fit of the $\Delta\mu$ XANES spectrum of $Co_{0.5}$ in $O_2$- or $N_2$-saturated electrolyte was obtained by the MXAN code with the following procedure: (1) Theoretical XANES spectrum calculation of the $CoN_4C_{12}$ moiety as reference, using the spectrum under nitrogen at 0.8 V. (2) At each step of the fitting procedure the program calculated the theoretical spectrum of a $CoN_4C_{12}$-$O_2$ moiety under oxygen. (3) The difference spectrum is calculated.

**DFT-D computation**. DFT-D computations were carried out on cluster models shown in Supplementary Fig. 6. The dangling bonds were saturated with hydrogen. The computational method is based on the DFT cluster approach augmented with an empirical dispersion term as implemented in deMon2k computer program used for the present computations[61]. For the exchange-correlation functional, the approximations of Perdew–Burke–Ernzerhof's 1996 was used[62]. Triple-ζ basis sets were used for the C and H atoms and double-ζ bases plus polarization for Co(II) cations[63]. Automatically generated auxiliary functions up to $l = 2$ (for the metal atom) and 3 (for H, C and O atoms) were used for fitting the density. A quasi-Newton method in internal redundant coordinates with analytical energy gradients was used for structure optimization. The convergence was based on the Cartesian gradient and displacement vectors with thresholds of $10^{-3}$ a.u. The energy convergence was set to $10^{-7}$ a.u. The binding energy of cobalt cation per Co–N bond (BE) and adsorption energy of end-on adsorbed $O_2$ molecule ($E_{ads}$) are obtained as

$$BE = E_{tot}[complex] - E_{tot}(Co(II)) - E_{tot}\left[\left(N_4C_y\right)^{2-}\right]. \qquad (3)$$

$$E_{ads} = E_{tot}[O_2/complex] - E[O_2] - E[complex]$$

In the above formula, the total energies for the ground state triplet dioxygen with total spin $S = 1$ (two unpaired electrons) and $Co^{2+}$ in its quartet ground state (total spin $S = 3/2$) were used. The relative energies $\Delta E$ with spin number are obtained as a difference between the total energies of the higher-energy minima states and the ground state. The atomic spin density was computed for all the minima energy structures using Mulliken[64] and Hirshfeld[65] population analysis. Both population schemes led to nearly identical results. Cobalt divalent cations were used. The relative energies $\Delta E$ with spin number are obtained as a difference between the total energies of the higher-energy minima states and the ground state.

**SQUID measurements**. The molar magnetic susceptibility $\chi_m$ was measured from 2 to 300 K for $Co_{0.5}$ and N–C with a Superconducting QUantum Interference Device (SQUID) (MPMS XL-7T, Quantum Design) at a magnetic field of 5,000 Oe. A mass of 16 mg of $Co_{0.5}$ or N–C was weighed and introduced in a polymer straw. The average effective magnetic moment of cobalt atoms ($\mu_{eff}$) was then obtained by fitting the plot of $1/\chi_m$ (in mol of cobalt atoms per emu) vs. $1/T$ with a linear law in the region 15–77 K. The value for the slope obtained is defined as a $1/C_m$. From the $C_m$-value, the average effective magnetic moment of cobalt atoms ($\mu_{eff}$) is directly calculated via the relation $\mu_{eff} = 2.82 \cdot C_m^{1/2}$, in units of Bohr magneton ($\mu_B$).

The average spin of cobalt ions is then obtained via $\mu_S = g \cdot \mu_B \cdot \text{sqrt}(s(s+1))$, where $g$ is equal to 2.002 and $s$ is the average spin density for all cobalt moieties present in $Co_{0.5}$.

**TEM measurements**. For transmission electron microscopy (TEM), the sample powders were dispersed in ethanol using an ultrasonic bath and deposited onto holey-carbon grids. TEM imaging and SAED were carried out on a JEM-2100HCKM microscope (JEOL Ltd., Japan) operating at 120 keV.

**Data availability**. The data that support the findings of this study are available from the corresponding authors upon reasonable request. The X-ray absorption Spectroscopy raw data associated with this work is permanently stored at SOLEIL and available upon request.

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

## Acknowledgements

This research was supported by ANR under contract 2011 CHEX 004 01 and by the Project 'PEMFC-SUDOE' -SOE1/P1/E0293 which is co-financed by the European Regional Development Fund in the framework of the Interreg Sudoe programme. We also acknowledge Synchrotron SOLEIL (Gif-sur Yvette, France) for provision of synchrotron radiation facilities at beamline SAMBA (proposal numbers 20120701 and 20150913), Valérie Briois for valuable discussions and Álvaro Reyes-Carmona (formerly at Université Montpellier, now at ICIQ, Spain) for sharing the benchmark IrO2 polarization curve.

## Author contributions

A.Z. participated in the XAS measurements and performed the XAS data analysis. N.R.-S. conducted the catalyst synthesis and electrochemical studies and participated in XAS measurements, T.M. conducted the DFT analysis. J.L. performed the SWV experiments. Q.J. and S.M. contributed to the data interpretation and manuscript writing. S.S., P.K. and E.F. participated in the XAS data acquisitions and contributed to the data

interpretation. G.F.H. and S.L. performed the TEM characterization. F.J. and A.Z. supervised the project, wrote and edited the manuscript, with inputs from all authors.

## Additional information

**Competing interests:** The authors declare no competing financial interests.

