## [Peer Review File · Nature Communications]

Editorial Note: Two figures in this peer review file have been reproduced with permissions from J. Phys. Chem. C 116 (2012) 16001, and Nature Materials 14 (2015) 937.

Reviewers' comments:

Reviewer #1 (Remarks to the Author):

The manuscript by Zitolo et al is an interesting complement to the article in Nature Materials 14, 2015. Mainly by means of spectra fitting, the authors seem to identify various active sites on Co-N-C materials for the oxygen reduction reaction, and the features of such sites are compared to similar Fe-N-C materials and the activities benchmarked with respect to Pt.

I think the manuscript is well written and thorough in the experimental part, but I should also say that the theoretical part makes big conclusions based on incomplete evidence. Thus, I believe that this article certainly deserves to be published in Nature Communications as theory is not its main focus and the calculations are only there to facilitate the discussion of the experimental findings. Nonetheless, the authors need to address the comments below:

Line 120-122: "Molecular cobalt catalysts are inherently less selective than iron ones, due to the more noble character of cobalt and hence weaker interaction with oxygen intermediates".

What is inherent to Co sites that makes them less selective? A weaker interaction with oxygen does not necessarily imply a decrease in selectivity. What is in reference 42 leading the authors to think so? Can they elaborate on that?

Lines 209-212: "This indicates that cobalt binds O₂ more weakly compared to iron-based moieties, which has deep consequences on the expected positions of Co- and Fe-N-C catalysts in a volcano plot for ORR (Figure 8 in Ref. 42) and also possibly on the rate-determining step for ORR".

I agree that Co binds O₂ more weakly than Fe, but that does not necessarily imply that (i) they will have different activities or (ii) they will have different rate-determining steps for the ORR. For (i), if the top of the volcano is located at an intermediate binding energy between Co and Fe, they can have the same activity. For (ii), if Co and Fe are located on the same side of the volcano plot, they will supposedly have the same limiting step.

The authors calculated some oxygen adsorption energies which clearly indicate, in line with the literature, that Co centers bind more weakly than Fe centers. However, they have not built a volcano plot with their sites and, therefore, cannot claim anything with regards to the activity of the sites or their limiting steps for the ORR.

Moreover, if the authors intend to extrapolate from the results of molecular catalysts in reference 42, they should be cautious, as the activities/adsorption energies of the MeN₄ centers are not necessarily the same for solid-state catalysts and molecular catalysts, because of different ligands, oxidation states, etc. This is shown in ref. 42 and by Garcia-Lastra and coworkers (Chem. Sci. 8, 2017) for various rings and ligands and has been shown by Koper and coworkers (Surf. Sci. 607, 2013) between porphyrins and solid-state materials with similar sites. The authors also seem to be aware of this, as in the introduction they quote ref. 21 to say that the differences between MeN₄C₁₂ and MeN₄C₁₀ can be up to 0.7 eV.

Line 217: "Hence, cobalt-moieties with the highest calculated O₂ adsorption energy should be the most active ones, which gives the following expected ORR turnover frequency ranking CoN₂C₅ > CoN₃C₁₀,porp > CoN₄C₁₂".

Again the same: the authors do not know what the optimal O₂ adsorption energy is so that they can claim that the moieties with the highest O₂ adsorption energies are the most active ones. The quoted claim would imply that all Co sites are located on the weak-binding side of the volcano plot, which is something they did not prove, so that strengthening the adsorption energies corresponds unambiguously to an activity increase.

Line 228-231: "According to the DFT spin density analysis and selecting the best candidates on the basis of EXAFS and XANES analysis (Supplementary Table S4), Co_{0.5} contains preferentially CoN₄C₁₂ and O₂-CoN₂C₅ in near 50/50 relative distribution (Saverage = 1.35 for a 50%/50% distribution of cobalt into CoN₄C₁₂ and O₂-CoN₂C₅ ground-state moieties)".

How do the authors conclude this? Is it simply an average between a spin state that is below the average and another one that is above it? If it is so, then many other possibilities exist when including other active sites in the average and changing their relative proportions.

Reviewer #2 (Remarks to the Author):

This is an interesting manuscript that looks to identify catalytic sites in cobalt-nitrogen carbon materials. To clarify the main findings and conclusions of the work, the authors are invited to make the following revisions:

1. The organization of the manuscript is not very clear. In the abstract, the authors say that the detailed structure of the catalyst was identified using XANES. In the main text, however, the authors use a combination of EXAFS, XANES, magnetic susceptibility, and DFT spin density analysis to identify the sites in their Co_{0.5} material (lines 228-231). Then in conclusions, the authors do not specify the used techniques or the identified sites, but generally claim that the manuscript identified detailed structures of single-atom catalytic sites. Furthermore, the future direction of research that the work helped defined is not clearly stated.
2. Further clarification is needed in the XANES analysis. Do the authors believe that there could be other structures that will reproduce the measured spectrum or do they think that only the three structures identified in the manuscript are possible candidates? Additionally, is the proposed composition of active sites on line 230 (through a combination of various techniques) a definitive identification or could the data be explained in a different manner?
3. In Figure 4, panels c and d, it would be helpful to keep the axis of the insets the same, in case the difference between O₂ and N₂ is desired to be emphasized.
4. The title includes hydrogen electrochemical reactions, but the manuscript does not have sufficient characterization to warrant the inclusion in the title. Furthermore, it is customary to record HER in H₂ atmosphere to ensure that hydrogen concentration at the surface of the catalyst doesn't change throughout the experiment (as the potential becomes more anodic). This is because a change in hydrogen surface concentration will change the onset potential for HER. In future work, the authors could do a more extensive characterization of hydrogen electrochemical reactions and identify which sites are active for HER and what sites are present when the catalyst is inactive for HOR.
5. Similarly, OER should be recorded in O₂ (to ensure defined conditions at the surface). In the OER section, the authors claim that the onset of OER in acidic medium corresponds to the redox peak at 1.25 V (line 311) but provide no evidence for this claim. The reported OER characterization starts at 1.3 V. Furthermore, the catalyst doesn't show significant activity until about 1.5-1.6 V. To warrant the claim, the authors should show the curve from at least 1.1 V and add the return sweep for both Co-N-C and N-C, so that the reader can get an understanding of the relative capacitive currents of the two materials.

(Note: in our responses below, all figure, table and reference numbers refer to those of the first submitted version, unless otherwise mentioned.)

Reviewers' comments:

Reviewer #1:

The manuscript by Zitolo et al is an interesting complement to the article in Nature Materials 14, 2015. Mainly by means of spectra fitting, the authors seem to identify various active sites on Co-N-C materials for the oxygen reduction reaction, and the features of such sites are compared to similar Fe-N-C materials and the activities benchmarked with respect to Pt. I think the manuscript is well written and thorough in the experimental part, but I should also say that the theoretical part makes big conclusions based on incomplete evidence. Thus, I believe that this article certainly deserves to be published in Nature Communications as theory is not its main focus and the calculations are only there to facilitate the discussion of the experimental findings.

We agree with the constructive criticism of the reviewer that calculated O₂ binding energies cannot fully describe and predict the ORR activity and selectivity of such molecular sites dispersed in a carbon matrix. The morphology of the hosting carbon matrix and atomic-level details (filling of the electronic orbitals etc) of the various cobalt-based sites (or iron-based sites) may also tune to some extent the turnover frequency and selectivity of such sites. On the other hand, the authors feel it is important to interpret in as much as possible the reactivity of such sites with general descriptors and catalysis concepts.

Action: In the revised manuscript, we have tried to be even clearer in distinguishing clear conclusions from proposed concepts that, while fitting with the experimental and/or theory results of the present work, would need additional data or samples to be definitely validated in the form of volcano plots.

Nonetheless, the authors need to address the comments below: Line 120-122: "Molecular cobalt catalysts are inherently less selective than iron ones, due to the more noble character of cobalt and hence weaker interaction with oxygen intermediates". What is inherent to Co sites that makes them less selective? A weaker interaction with oxygen does not necessarily imply a decrease in selectivity. What is in reference 42 leading the authors to think so? Can they elaborate on that?

At a same oxidation state, cobalt has more d-electrons than iron. Free cobalt (non-coordinated) also shows a higher redox potential than free iron, which can be translated in general words as generally "being more noble" than iron. Figure 3 in Ref. 42 shows a clear linear correlation between the formal potential E° for Metal^{III}/Metal^{II} in porphyrins and phthalocyanines (experimentally measured) and their DFT-calculated O₂ binding energy. Thus, for a given macrocyclic structure (e.g. phthalocyanine, labelled Pc in that figure), the experimental formal potential for the metals in Pc follow the trend for free metal ions, with $E^\circ(\text{Co}^{\text{III}}/\text{Co}^{\text{II}}) > E^\circ(\text{Fe}^{\text{III}}/\text{Fe}^{\text{II}}) > E^\circ(\text{Cr}^{\text{III}}/\text{Cr}^{\text{II}})$. This however does not impede in particular cases some CoN₄ macrocycles to show a slightly lower formal potential than other Fe macrocycles, if one changes both the nature of the metal and the macrocyclic structure (only one such exception can be found in Fig. 3 of Ref. 42: $E^\circ(\text{Co}^{\text{III}}/\text{Co}^{\text{II}})$ in 4(Ph)CoP < $E^\circ(\text{Fe}^{\text{III}}/\text{Fe}^{\text{II}})$ in FeTPyPz).

Thus, CoN₄ macrocycles have generally a higher formal potential than FeN₄ macrocycles and also a weaker binding energy for O₂. Due to scaling relationships, a weaker binding energy for O₂ also implies a weaker binding energy for H₂O₂ (Fig. 4 in Ref. 42: Angew Chemie Int Ed 55 (2016) 14510). The weaker binding energy for H₂O₂ in turn can explain why the selectivity to water formation is less for cobalt than for iron sites. Indeed, numerous experimental reports have shown that cobalt-N₄ macrocycles (at least, non-pyrolyzed) are less selective than iron-N₄ macrocycles.

Action: the sentence on line 120-122 in the first version has been modified to clarify it. The new text reads: "By analogy with what is known for unpyrolyzed Metal-N₄ macrocycles, this result may be explained by the generally higher formal potential of Metal^{III}/Metal^{II} for cobalt vs. iron macrocycles, leading to weaker interaction with oxygen intermediates, including H₂O₂.⁴² Square-wave-voltammetry in acidic medium supports this possible explanation, identifying redox-peak positions at 1.25 V and 0.75 V vs. RHE for Co_{0.5} and Fe_{0.5}, respectively"

Lines 209-212: "This indicates that cobalt binds O₂ more weakly compared to iron-based moieties, which has deep consequences on the expected positions of Co- and Fe-N-C catalysts in a volcano plot for ORR (Figure 8 in Ref. 42) and also possibly on the rate-determining step for ORR". I agree that Co binds O₂ more weakly than Fe, but that does not necessarily imply that (i) they will have different activities or (ii) they will have different rate-determining steps for the ORR. For (i), if the top of the volcano is located at an intermediate binding energy between Co and Fe, they can have the same activity. For (ii), if Co and Fe are located on the same side of the volcano plot, they will supposedly have the same limiting step.

We agree with the reviewer that, without any *a priori* constraints, the above mentioned possibilities of positions of Co-N-C catalysts and Fe-N-C catalysts in a volcano plot are indeed possible, and some of them could explain our experimental results. On i) first, since it is experimentally observed that Co_{0.5} is less active than Fe_{0.5} (Figure 2d), the theoretical possibility of identical ORR activity for two completely different O₂ BE symmetrically positioned around the apex of the volcano plot cannot apply in our case. Fe_{0.5} must thus be situated closer to the apex than Co_{0.5}. However, we agree with the reviewer that we do not know yet whether Fe_{0.5} is on the weak binding side or on the strong binding side.

[Redacted]

Action: the sentence has been modified for clarifying these different possibilities. The new text reads: "Due to the lower experimental ORR activity of Co_{0.5} vs. Fe_{0.5} (**Figure 2d**), one can unambiguously conclude that cobalt moieties are situated on the weak-binding side of a volcano plot. While no definitive conclusion can be made on the volcano-plot position of iron moieties in Fe_{0.5} only from the present work (the weak and strong binding branches being both possible), yet unpublished work on Fe_{0.5} post-treated with H₂O₂ suggests that Fe_{0.5} is also positioned on the weak binding side of a volcano plot."

We also discuss the O₂ BE of Co and Fe in the revised conclusion section.

The authors calculated some oxygen adsorption energies which clearly indicate, in line with the literature, that Co centers bind more weakly than Fe centers. However, they have not built a volcano plot with their sites and, therefore, cannot claim anything with regards to the activity of the sites or their limiting steps for the ORR.

As the reviewer proposed in the previous question, and assuming a volcano plot exists for such Metal-N_x sites, it is possible to rationally discuss from the knowledge of the activity and binding energy of only two catalysts the possible position in the volcano plot of these two catalysts. From the combined experimental data and DFT, it follows that the cobalt catalyst must sit on the weak binding side, while for the iron catalyst, both possibilities are still open. We believe this has now been clarified by our action to the previous question.

Regarding that no claim can be made as to the activity of the sites, we do not exactly understand the statement. The two catalysts here contain only atomically dispersed Metal-N_x sites and at a same metal content, so that we can quite safely assume a similar number of active sites in both catalysts. The activity of the sites can thus be compared by comparing the ORR activity measured at 0.8 V vs RHE. Regarding the ORR limiting step, we actually do not discuss this in the paper, and only discuss the selectivity on the basis of the experimentally measured %H₂O₂ during ORR.

Moreover, if the authors intend to extrapolate from the results of molecular catalysts in reference 42, they should be cautious, as the activities/adsorption energies of the MeN₄ centers are not necessarily the same for solid-state catalysts and molecular catalysts, because of different ligands, oxidation states, etc. This is shown in ref. 42 and by Garcia-Lastra and coworkers (Chem. Sci. 8, 2017) for various rings and ligands and has been shown by Koper and coworkers (Surf. Sci. 607, 2013) between porphyrins and solid-state materials with similar sites. The authors also seem to be aware of this, as in the introduction they quote ref. 21 to say that the differences between MeN₄C₁₂ and MeN₄C₁₀ can be up to 0.7 eV.

We fully agree that a fundamental difference exists between unpyrolyzed MeN₄ macrocycles simply adsorbed on carbon and the MeN_xC_y type moieties existing in pyrolyzed materials and that are covalently integrated in (disordered) graphene sheets. Even for a same coordination environment in the two first coordination spheres (e.g. CoN₄C₁₂ in pyrolyzed material and in porphyrins), the electron density at the metal center and hence the O₂ binding energy could differ due to different longer-range environments. We believe this is the main reason why pyrolyzed Me-N-C materials are much more stable in acid medium and more active toward ORR than adsorbed MeN₄ macrocycles on carbon. The O₂ BE calculated by us for CoN₄C₁₂ is -0.80 eV (Table S4), which seems to be larger (in absolute value) than the O₂ BE calculated for any Co-porphyrin or phthalocyanines (Fig. 8 in Ref. 42, range of -0.6 to -0.35 eV). Of course, also the optimum O₂ adsorption energy is probably different from adsorbed macrocycles to graphene-integrated MeN_xC_y moieties...The learnings from reactivity of molecular catalysts cannot thus be blankly applied to pyrolyzed Metal-N-C catalysts, but the reactivity concepts still apply, and to some extent, the main trends also (Co and Fe remaining the best metal centers for ORR, both for molecular and graphene-integrated sites, as experimentally observed and also theoretically observed in Surf Sci 607 (2013) 47)

Line 217: "Hence, cobalt-moieties with the highest calculated O₂ adsorption energy should be the most active ones, which gives the following expected ORR turnover frequency ranking CoN₂C₅ > CoN₃C_{10,porp} > CoN₄C₁₂".

Again the same: the authors do not know what the optimal O₂ adsorption energy is so that they can claim that the moieties with the highest O₂ adsorption energies are the most active

ones. The quoted claim would imply that all Co sites are located on the weak-binding side of the volcano plot, which is something they did not prove, so that strengthening the adsorption energies corresponds unambiguously to an activity increase.

While we agree that we do not know what the optimal O₂ adsorption energy is, the much lower calculated O₂ BE for all CoN_xC_y moieties that are good site candidates (from a XANES point of view) (CoN₄C₁₂, CoN₃C_{10,porp}, and CoN₂C₅ having -0.80, -1.23 and -1.26 eV O₂ BE) than FeN₄C₁₂ (-1.84 eV O₂ BE) and the parallel lower overall experimental ORR activity of Co_{0.5} vs. Fe_{0.5} clearly places cobalt moieties on the weak binding leg of a volcano plot (and, as discussed in a reply to one of the previous questions of the reviewer, we do not know where the Fe sites are located in the volcano plot, and the activity ranking of various FeN_xC_y moieties is thus not discussed).

Action: we have added a clarification in the sentence, underlined below: "The O₂ adsorption energy of molecular cobalt catalysts is often weaker than desired for an ideal metal-centered moiety,⁴² and the same conclusion is made from the present combined experimental and theoretical study on the pyrolyzed Co_{0.5} catalyst. Hence, cobalt-moieties with the highest calculated O₂ adsorption energy should be the most active ones..."

Line 228-231: "According to the DFT spin density analysis and selecting the best candidates on the basis of EXAFS and XANES analysis (Supplementary Table S4), Co_{0.5} contains preferentially CoN₄C₁₂ and O₂-CoN₂C₅ in near 50/50 relative distribution (Saverage = 1.35 for a 50%/50% distribution of cobalt into CoN₄C₁₂ and O₂-CoN₂C₅ ground-state moieties)". How do the authors conclude this? Is it simply an average between a spin state that is below the average and another one that is above it? If it is so, then many other possibilities exist when including other active sites in the average and changing their relative proportions.

Yes, the interpretation above is correct and we reported the 50/50 relative distribution as an estimate yielding an apparent spin density that is close to the one that is derived from the magnetic measurement. Looking at Table S4 (bold numbers corresponding to most stable configurations), knowing that the average experimental spin on cobalt is 1.33 and that the CoN₄C₁₀ moiety cannot be present in a large fraction because it does not match the experimental XANES spectrum, the possibilities to obtain the experimental effective spin by averaging using different proportions of CoN_xC_y sites is quite restrained, if one looks only at the ground state (highlighted in bold) moieties whose calculated XANES can match the experimental XANES: the three possible spins on cobalt are 0.88, 0.58 and 1.83.

The maximum fraction of cobalt sites corresponding to the highest spin 1.83 to result in an average spin of 1.33 is thus 60% (the remainder 40% being then only the cobalt site with the lowest spin: 0.58). The lowest fraction of cobalt sites corresponding to the spin 1.83 can also be deduced to be ca 47% (the remainder 53% being then only the site with medium spin, 0.88).

In conclusion, the fraction of site with highest spin 1.83 (CoN₂C₅ or O₂-CoN₂C₅) must be in the range 47-60%, while the fraction of the other two sites can vary from ca 0% to 40-53%. For example, fixing a fraction of 53% of CoN₂C₅ site, one can find that the fraction of the other two sites must be ca 16 % for CoN₃C₁₀ and 31% for CoN₄C₁₂ to result in an effective spin of 1.33.

Action: a supplementary Table S5 with the three cases of combinations of fractions of the three sites discussed above and leading to an average spin 1.33 has been added.

Fraction of O ₂ - CoN ₂ C ₅ sites (spin 1.83)	Fraction of CoN ₃ C _{10,porp} sites (spin 0.58)	Fraction of CoN ₄ C ₁₂ sites (spin 0.88)	Average spin
60	40	0	1.33
47	0	53	1.33
53	16	31	1.33

The added text in the main manuscript reads:

“According to the DFT spin density analysis and selecting the three best candidates on the basis of EXAFS and XANES analysis (**Supplementary Table S4**, CoN₄C₁₂, CoN₃C_{10,porp} and O₂-CoN₂C₅ in their ground-state, spin density of cobalt of 0.88, 0.58 and 1.83, respectively), one can propose that Co_{0.5} contains for example CoN₄C₁₂ and O₂-CoN₂C₅ in a 53% / 47% distribution, or CoN₃C_{10,porp} and O₂-CoN₂C₅ in a 40% / 60% distribution. Both distributions result in a theoretical s_{average} value of 1.33, similar to the experimental value (**Supplementary Table S5**). Considering only these three motifs, the fraction of O₂-CoN₂C₅ sites must be in the range of 47-60%, the remainder being split between CoN₄C₁₂ and CoN₃C_{10,porp} sites.”

Reviewer #2:

This is an interesting manuscript that looks to identify catalytic sites in cobalt-nitrogen carbon materials. To clarify the main findings and conclusions of the work, the authors are invited to make the following revisions:

1. The organization of the manuscript is not very clear. In the abstract, the authors say that the detailed structure of the catalyst was identified using XANES. In the main text, however, the authors use a combination of EXAFS, XANES, magnetic susceptibility, and DFT spin density analysis to identify the sites in their Co_{0.5} material (lines 228-231). Then in conclusions, the authors do not specify the used techniques or the identified sites, but generally claim that the manuscript identified detailed structures of single-atom catalytic sites. Furthermore, the future direction of research that the work helped defined is not clearly stated.

We agree that not only XANES was involved in the detailed structural and electronic identification of the likeliest cobalt sites in the present material.

Action: That part of the abstract has been modified (underlined words) to: "Here, we pyrolytically synthesized a Co-N-C material comprising only atomically-dispersed cobalt ions and identified with X-ray absorption near-edge spectroscopy (EXAFS and XANES), magnetic susceptibility measurements and DFT analysis the detailed structure and electronic state of three porphyrinic moieties, CoN₄C₁₂, CoN₃C_{10,porp} and CoN₂C₅."

The conclusion was similarly modified, and we also modified the future possible directions that this work helped identify.

2. Further clarification is needed in the XANES analysis. Do the authors believe that there could be other structures that will reproduce the measured spectrum or do they think that only the three structures identified in the manuscript are possible candidates? Additionally, is the proposed composition of active sites on line 230 (through a combination of various techniques) a definitive identification or could the data be explained in a different manner?

We believe that most CoN_x planar or near-planar structures have been investigated in the present work, on the basis of a systematic approach. Also, the single atom defect structure (replacing one C atom by Co and replacing the 3 nearest C by 3 N atoms) did not match the XANES experimental spectrum and had in addition low stabilization energy. There are some constraints that are applied to the spatial position of the cobalt and light element atoms in the present XANES analyses of course. Highly non-planar arrangements of cobalt ions with N or C atoms (near tetrahedral coordination for example) have not been investigated per se. However, the planar CoN_x models with O₂ adsorbed on top lead, to some extent, to such model sites with small out-of-plane displacement of cobalt in some cases. A site geometry that correctly reproduces the XANES experimental spectrum is thus a necessary requirement for the active site assignment, but this does not preclude the existence of other structures. The combination of EXAFS and XANES analysis reduces however the number of possible structures.

Composition of active sites: In our approach, we tried to match the XANES experimental spectrum of that "model" Co-N-C catalyst (no metallic cobalt) every time only with a single CoN_xC_y moiety (not a mix of different CoN_xC_y moieties in a single hypothetical catalyst). The present work identifies three likeliest cobalt moieties (CoN₄C₁₂, CoN₃C_{10,porp} and O₂-CoN₂C₅),

due to their proper match of calculated XANES to experimental XANES. In addition, the comparison of the spin density of cobalt from experiment and DFT calculation clearly indicates that a significant fraction of cobalt must be in the CoN_2C_5 configuration (calculated spin 1.83) to explain the average spin density measured (1.33), because all other spin values calculated for other CoN_xC_y moieties are only 0.88 or even lower (Table S4). The proposed composition on line 230 is however only a rough estimate.

The same question was indeed also asked by reviewer 1 (see his/her last question and our response to that comment).

Action: a supplementary Table S5 with three cases of combinations of fractions of the three sites discussed above and leading to an average spin 1.33 has been added.

Fraction of O_2 - CoN_2C_5 sites (spin 1.83)	Fraction of $\text{CoN}_3\text{C}_{10,\text{porp}}$ sites (spin 0.58)	Fraction of $\text{CoN}_4\text{C}_{12}$ sites (spin 0.88)	Average spin
60	40	0	1.33
47	0	53	1.33
53	16	31	1.33

The added text in the main manuscript reads:

“According to the DFT spin density analysis and selecting the three best candidates on the basis of EXAFS and XANES analysis (**Supplementary Table S4**, $\text{CoN}_4\text{C}_{12}$, $\text{CoN}_3\text{C}_{10,\text{porp}}$ and O_2 - CoN_2C_5 in their ground-state, spin density of cobalt of 0.88, 0.58 and 1.83, respectively), one can propose that $\text{Co}_{0.5}$ contains for example $\text{CoN}_4\text{C}_{12}$ and O_2 - CoN_2C_5 in a 53% / 47% distribution, or $\text{CoN}_3\text{C}_{10,\text{porp}}$ and O_2 - CoN_2C_5 in a 40% / 60% distribution. Both distributions result in a theoretical s_{average} value of 1.33, similar to the experimental value (**Supplementary Table S5**). Considering only these three motifs, the fraction of O_2 - CoN_2C_5 sites must be in the range of 47-60%, the remainder being split between $\text{CoN}_4\text{C}_{12}$ and $\text{CoN}_3\text{C}_{10,\text{porp}}$ sites.”

3. In Figure 4, panels c and d, it would be helpful to keep the axis of the insets the same, in case the difference between O_2 and N_2 is desired to be emphasized.

Action: This is an interesting suggestion to improve the figure, and we now have modified insets c) and d) in figure 4 accordingly.

4. The title includes hydrogen electrochemical reactions, but the manuscript does not have sufficient characterization to warrant the inclusion in the title.

Action: the title has been modified to "Identification of Catalytic Sites in Cobalt-Nitrogen-Carbon Materials for the Oxygen Reduction Reaction"

Furthermore, it customary to record HER in H_2 atmosphere to ensure that hydrogen concentration at the surface of the catalyst doesn't change throughout the experiment (as the potential becomes more anodic). This is because a change in hydrogen surface concentration will change the onset potential for HER.

Yes, we agree with that aspect (H_2 saturated electrolyte) highlighted by the reviewer.

Action: we have re-measured the HER curves in H₂-saturated acid solution, and the new curve is seen in a new figure in supporting information (the HER and OER data has been removed from the main manuscript)

In future work, the authors could do a more extensive characterization of hydrogen electrochemical reactions and identify which sites are active for HER and what sites are present when the catalyst is inactive for HOR.

Non-PGM catalysts for HER that are much more active than the present one seem to be those comprising metallic particles (e.g. Fe or Co) embedded in thin graphite shells (e.g. Tavakkoli et al, *Angew. Chemie* 54 (2015) 4535). Some recent papers have also reported fairly high HER activity for molecular pyrolyzed Co-N-C catalysts (e.g. *Nature Commun.* 6 (2015) 7992), which prompted us to evaluate the HER activity of the present catalyst, but we find a much lower HER activity here. In contrast, some Metal-N-C catalysts derived from ZIF-8 and metal acetates comprising metallic particles embedded in carbon (including Co-N-C) evaluated at our laboratory worked quite well for HER (*J. Electrochem. Soc.* 162 (2015) H719).

Regarding HOR, we are not aware of any pyrolyzed Metal-N-C catalyst that shows any activity toward HOR. We have verified this from time to time for different type of Co-N-C catalysts, including the present one. We cannot explain however why HER is catalyzed but HOR not.

5. Similarly, OER should be recorded in O₂ (to ensure defined conditions at the surface). In the OER section, the authors claim that the onset of OER in acidic medium corresponds to the redox peak at 1.25 V (line 311) but provide no evidence for this claim. The reported OER characterization starts at 1.3 V. Furthermore, the catalyst doesn't show significant activity until about 1.5-1.6 V. To warrant the claim, the authors should show the curve from at least 1.1 V and add the return sweep for both Co-N-C and N-C, so that the reader can get an understanding of the relative capacitive currents of the two materials.

The redox peak for Co was reported in Fig. S2 of the first submitted version to be *ca* 1.25 V vs RHE (measured in acid medium). At this potential one expects part of the cobalt ions to be in +III oxidation state and therefore the onset for OER should be close to that potential. Figure 5b (OER in acid) shows, to the opinion of the authors, that the OER onset is very well aligned with the redox peak position. In addition, the capacitive current cannot account for the difference Co_{0.5} and N-C because both samples have similar BET area and carbon microstructures.

Action:

a) we have modified the x-axis scale in the former figure 5a-b to include 1.25 V and better show the curve onset for OER (now, **figure S10**).

b) The OER data has been re-measured in O₂-saturated electrolyte. The OER data in acid medium is now shown only in the supporting information.

c) The information that both N-C and Co_{0.5} have similar BET area and hence capacitive currents has been added in the new **figure S10**

d) we have added operando XANES at various OER potentials, **figure S11**.

Reviewer #3:

In this work a detailed XAS analysis of a Co_{0.5} catalyst is performed based on ex-situ data and in-situ measurements performed in oxygen and nitrogen saturated electrolyte out of the potential range for oxygen reduction reaction. In addition to this, R(R)DE and fuel cell tests as measures for oxygen reduction activity were performed as well as tests on HOR, HER and OER.

DFT calculations of energies of formation for different CoN_xC_y sites (with and without oxygen ligands) were made in order to verify the formation probability of the different species. In addition, SQUID measurements were made to determine the average ground state spin occupation of the Co_{0.5} catalyst. For reasons of comparison a similarly prepared Fe_{0.5} catalyst, N-C catalyst and Pt/C are investigated electrochemically. The main conclusion based on these data is that it might be any CoN_xC_y moiety that catalysis the ORR in acidic, there is no information on HER, HOR or OER or ORR in alkaline.

Concerns:

Operando XANES data clearly indicate that there are significant potential-dependent changes observable for Fe_{0.5}. This is explained by the redox peak of iron at 0.75V. It is important to notice that the XANES signature changes continuously and not only within the range of the redox peak. In contrast, all measurements in N₂ saturated H₂SO₄ on Co_{0.5} give identical XANES signatures in the potential range 0.0 to 1.0 V.

Why is there such a strong change visible for iron, even though the oxidation state changes only at 0.75 V? (By similarity 0.5 V and 0.3 V are similarly far away from the redox peak (0.75 V) in case of iron compared to 1.0 and 0.8 V vs. redox peak (1.25 V) in case of cobalt) It seems that the change in oxidation state cannot be the only explanation for the behavior of iron vs cobalt.

The reviewer is absolutely correct that the XANES spectra for Fe_{0.5} seem to continue changing even well below the redox peak position (no redox signal visible in Fig. S2 below 0.65 V). Above 0.7-0.8 V there is no change, in agreement with the fading redox signal (Fig. S2 in the first submitted version). As suggested by the reviewer, there might be more than only a redox change that tunes the XANES potential dependence in Fe_{0.5}.

This possibility was in fact mentioned in the first submitted version:

Quoted from first submitted version : "The spectral changes observed in **Figure 4b** probably result not only from a change in oxidation state, which would *a priori* only shift the edge position of the XANES spectra, but also from deeper structural changes and reorganisation of the N (or C) ligands, as proposed in Ref.²⁶ and/or an Fe(II) low-to-high spin-crossover.⁴⁹"

Action: we have added the following sentence for highlighting this unexplained fact: "It is however surprising that the XANES spectra of Fe_{0.5} are still changing below 0.6 V vs. RHE, while most Fe ions should already be in +II oxidation state. Spin-state or conformation changes may still be occurring below 0.6 V."

In addition: For Co_{0.5} the interaction with oxygen is similar at 0.8 V and 0.2 V whereas the current density is changing. This is assigned to a weak interaction of Co with oxygen (compared to Fe_{0.5}) and therefore low performance. However, if a Co moiety is the active site, I would assume a stronger interaction with oxygen for higher overpotentials. In contrast, for iron-based catalysts, with strong interaction with O₂ one would assume strong changes at

0.8 V for the XANES signature in N₂ vs. O₂ saturated electrolyte, as active sites should already adsorb oxygen but not reduce it to water.

Because Co does not change oxidation state between 0.8 and 0.2 V vs RHE (and no other change observed from XANES either) one does not expect a different O₂ adsorption on Co at these two potentials. This is in line with Fig. 4c, inset. The O₂ adsorption on cobalt sites is only the first step in ORR. If there is no driving force to electroreduce O₂, there will be no ORR, even if weak adsorption occurs. The fact that more current is produced at 0.2 vs. 0.8 V is only related to the potential of the electrons. It however remains clear that, at fixed potential (i.e. comparing ORR at a same potential for the electrons), the cobalt moiety binds O₂ more weakly than the Fe moiety does, and this impacts the ORR turnover frequency at fixed electrochemical potential.

In addition, the fitted ex-situ data always give better R² values for the structures with O₂ in end-on configuration.

Differences of 0.04 between the R_{sq}-values of CoN₄C₁₂ and O₂-CoN₄C₁₂, or of 0.11 between the R_{sq} values of CoN₃C_{10,porp} and O₂-CoN₃C_{10,porp}, are not significant. All three CoN_x porphyrinic moieties are possible active site candidates (with or without O₂ adsorbed, except for CoN₂C₅ that needs an axial Oxygen to fit the XANES spectrum), with low R_{sq} value near 1 (Table 1, rows Nbr 1-4, 6). The XANES analysis clearly excludes however that a large fraction of the cobalt sites is based on the pyridinic type moieties, for which the R_{sq} values is much higher, 2.6-3.2 (Table S1).

- To what extent do the fittings of XANES data in N₂ vs O₂ saturated electrolyte at 0.2 V vs 0.8 V indeed differ (four samples, the three proposed “active” moieties should be considered in the fittings with and without oxygen interaction → Six data sets per sample that should indicate for which of the species there is indeed most likely a change in the coordination environment N₂ vs O₂)

MXAN calculations of cobalt moieties in N₂ and O₂ saturated electrolytes are not really able to interpret the small differences of the experimental spectra occurring in correspondence with the white line. For this reason, we resorted to the $\Delta\mu$ technique to elucidate the origin of these spectral features. $\Delta\mu$ is a subtracting technique consisting in the difference given by the spectrum in the presence of an adsorbate and the spectrum without adsorbate. For this very reason, it is possible to better isolate the contribution of the adsorbing oxygen. The analysis works equally for the CoN₄C₁₂ and CoN₃C₁₀ moieties (the CoN₂C₅ structure has not been contemplated for this analysis, because its stable form exists only with a bonded oxygen).

- All operando data were measured in 0.5M H₂SO₄. How and on what sites the catalytic reactions proceed in alkaline is therefore out of the focus of this paper. Therefore, electrochemical data should be given for acidic electrolyte, only.

We agree with the reviewer. *Operando* data in alkaline were partly acquired but practical issues and limited beamtime impeded us collecting all the necessary data during a single beamtime run.

Action: all electrochemical data in alkaline electrolyte were removed.

- There are no operando data available for OER or HER conditions. In addition, the performance of the Co_{0.5} for both reactions is poor. Based on the provided data there is no link between HER, OER and XAS provided. Hence, the title about identification of HER and OER active sites is misleading and not justified by the data. OER, HOR and HER data should be omitted.

We agree with the reviewer that the OER and HER activities are rather poor and that the paper focuses on the ORR activity.

Action: The title of the paper has been modified to reflect that, and the data in alkaline electrolyte have all been removed. We however decided to keep the HER and OER data in acid medium, but those have been shifted to the supporting information. To demonstrate that the CoN_xC_y moieties are still present in OER conditions, we have now also added a supporting figure (**figure S11** in revised version) showing the XANES spectra at high potential, showing only a small difference to the spectra recorded near 1 V vs RHE. We are therefore confident that, during short time measurement and at room temperature (where carbon corrosion rate is sufficiently slow), the OER current observed was catalyzed by the same CoN_xC_y moieties whose structure was identified for the ORR. The HER conditions should definitely not lead to a destruction of such active sites, and we can also be confident that the CoN_xC_y moieties whose structure was identified for the ORR are responsible for the HER catalysis by that material.

Minor concerns:

TEM images (low magnification and high magnification) should be provided to proof the atomically dispersion of Co atoms.

Action: Low and high resolution TEM images have been obtained in collaboration with Stephen Lyth and George Harrington (Kyushu University, Japan). No cobalt particles could be observed in the images and no rings that can be assigned to metallic cobalt could be seen in the SAED patterns, confirming the EXAFS analysis. A short text describing this has been added in the main text, Results section, sub-section now entitled "Evidencing with *ex situ* EXAFS and TEM the absence of Co-Co bonds in Co_{0.5}". A new supplementary figure Fig. S1 shows the SAED pattern and low and high magnification images for an amorphous carbon domain and for a domain with graphitic sheets.

It is well known that H₂O₂ selectivity should be measured at low catalyst loading e.g. 100 μg cm⁻².

Action: We have now performed also the RRDE analysis at 200 μg cm⁻². The %H₂O₂ measured at 1600 rpm at this new loading is now shown in the modified figure 1. The %H₂O₂ is slightly higher than at 800 μg cm⁻² (up to 11 % instead of 5% at higher loading).

If "ORR activity levels of at 2-3 wt% metal loading in the precursor (ref. 46)" (line 145), why are there several good performing catalysts available prepared from significantly higher iron loadings in the precursor (e.g. Wu et al., Science 2011)? I guess ref. 46 is not appropriate in this context.

We do not fully understand the detailed reasoning behind the reviewer's question: "why are there several good performing catalysts....higher iron loading in the precursors?"

We agree with the reviewer that several Fe- or Co-N-C catalysts with good ORR activity per total mass of catalyst (mass of carbon, nitrogen and metal) but having much higher Fe or Co content that used here have been reported, but this is perfectly compatible with the leveling-off of ORR activity above a certain threshold of metal content. If the ORR activity of a given Fe-N-C catalyst with a lot of iron is similar to that of another Fe-N-C catalyst with less iron, this might be explained by the fact that, in one catalyst, not only active Fe sites exist but also Fe structures that are ORR inactive.

Regarding the Fe-N-C catalysts reported in Wu et al., Science 332 (2011) 443 (Zelenay's group), Mössbauer spectra of the catalysts reported by the same authors in 2012 has clearly shown that the majority of Fe in such catalysts was ORR-inactive (J. Phys. Chem. C 116 (2012) 16001). In contrast, Mössbauer spectroscopy of Fe0.5 has shown that all Fe was coordinated as FeN₄ type moieties (Nature Materials 14 (2015) 937). To highlight this, the Mössbauer spectra are shown below.

Figure 4. Mössbauer spectrum of PANI-Fe-C_HT1_AL1_HT2 (after the first heat treatment at 900 °C, acid leach, and the second heat treatment at 900 °C) measured at room temperature. Final catalyst loading is 3 wt % Fe.

Above: Mössbauer spectrum for a catalyst prepared at Zelenay's group, Reproduced with permission from J. Phys. Chem. C 116 (2012) 16001 - final Fe loading: 3 wt % Fe; the ORR active FeN₄ moieties are represented by the doublets D1, D2, D3 (minor fraction of Fe signal)

Above: Mössbauer spectrum of the present Fe-N-C catalyst, $\text{Fe}_{0.5}$, prepared identically as the present Co-N-C catalyst $\text{Co}_{0.5}$, Reproduced with permission from Figure 1 in Nature Materials 14 (2015) 937. The ORR active doublets D1 and D2 represent 100% of Fe. Total Fe content 1.5 wt %

1187: “When O_2 is adsorbed end-on without out-of-plane displacement of cobalt”. Why does Co remain in plane?

This aspect is linked to the overall view of the absorption spectrum on the different active sites: we identified a number of possible structures existing in a planar configuration, $\text{CoN}_4\text{C}_{12}$, $\text{CoN}_3\text{C}_{10}$ and $\text{O}_2\text{-CoN}_2\text{C}_5$, together with the same but oxygen-bound moieties where the metal center may eventually leave the plane. Because of the weak binding of cobalt with oxygen, we believe that if an out-of-plane displacement occurs, it should be small.

11228-231: Based on the data it is suggested that the (ex-situ) catalyst contains $\text{CoN}_4\text{C}_{12}$ and $\text{O}_2\text{-CoN}_2\text{C}_5$ moieties. That means there is already a strong interaction with oxygen for that site that leads to the highest TOF based on the author’s suggestions (l. 217). Does this interaction with O_2 not contradict the operando XANES data?

To the authors opinion, there is no contradiction, because also the ability of the metal ions to “activate water” must be taken into account in interpreting the O_2 vs N_2 differential spectra: Fe is strongly binding O_2 but also O from H_2O , and therefore no difference is seen in N_2 - or O_2 -saturated aqueous electrolyte (fig. 4d). Co is weakly binding O_2 , and probably not binding O from H_2O at all. For this reason, a difference is seen between O_2 -saturated aqueous electrolyte (weak O_2 adsorption) and N_2 -saturated aqueous electrolyte (no axial oxygen atom) (fig. 4c).

The O_2 adsorption energy, even on CoN_2C_5 , is only -1.26 eV (Table S4). This is much lower than the O_2 adsorption energy calculated by us for the $\text{FeN}_4\text{C}_{10}$ and $\text{FeN}_4\text{C}_{12}$ moieties, -1.83 eV (Nature Materials 14 (2015) 937). It explains why it is less active toward ORR.

Number of minor ticks in Figure 3 (x-axis) is not useful.

For this Figure it would also be nicer to have the original data in the front and fitted data as background for better comparison.

We agree with the reviewer’s observation concerning the number of ticks and the data presentation.

Action: The number of ticks has been reduced in Figure 3 and in all the Supporting Figures in order to have 10 eV between two minor ticks. The experimental and theoretical spectra representation has been changed in Figure 3 and in all the Supporting Figures of the manuscript.

What energy was used as E0 (7.7093 KeV?).

The edge position E0 is a fitted parameter that differs from the value of metallic cobalt (7709 eV). For the five structures able to reproduce the experimental XANES spectrum, the E0 values are (in eV, errors are around ± 0.3):

CoN4C12 = 7717.36

O2-CoN4C12 = 7717.14

CoN3C10, porp = 7717.58

O2-CoN3C10, porp = 7717.80

O2-CoN2C5 = 7717.54

The E0 value of 7717.58 eV found for Co(II)Pc further demonstrates the +2 oxidation state of the cobalt in the ex-situ Co0.5 catalyst.

Figure 4 c-d: Please provide for Co0.5 and Fe0.5 four current density data obtained during the XANES measurements at 0.2 V and 0.8 V in O2 vs. N2 saturated electrolyte.

The requested data is in the author's opinion not fully relevant because the XANES operando cell setup does not allow fast mass-transfer of O₂, and thus a semi-infinite diffusion of O₂ from bulk electrolyte to the electrode surface takes place, resulting in a low ORR current (limited by mass transport, not by electro-kinetics).

The requested data would be relevant however for a cell setup with O₂ mass transport similar to or faster than that found in rotating disk electrode. We are planning on implementing a modified PEM fuel cell setup for operando XANES measurements with fast O₂ transport, but this will take 1-2 more years for completion.

Figure S1: Graphs are too similar in color and form. The trends are not easily recognizable.

The data in alkaline medium has now been removed (former Fig. S1a) and the data in acid medium has now been integrated in the modified figure 2.

Figure S9: HOR activity is measured by LSV between -0.1V to 0.4 V: All other conditions are missing in the methods section (catalyst loading, rotation? sweep rate? gas saturation?)

The conditions were 0.8 mg cm⁻² catalyst loading, 10 mV s⁻¹, 1600 rpm, H₂ saturated 0.1 M H₂SO₄. This information is now shown in the corresponding figure caption in supplementary information.

Methods section:

Catalyst preparation:

The ball milling conditions are not precise: what volume has the crucible, how many balls? Which diameter etc.

We omitted in the first version the details on the ballmilling procedure since they were included in several of our recent publications, but other aspects of the synthesis have in fact also been modified compared to our earlier works. We therefore have now included all ballmilling conditions in the Methods section. The equipment, ZrO₂ ball diameters and number, and crucible volume are indicated.

Action: the requested information has been added in Methods, and the new information is underlined in the text below.

Weighed amounts of the powders of Co(II)Ac (15.97 mg), phen (200 mg) and ZIF-8 (800 mg) were ball-milled (Pulverisette 7 premium, Fritsch) at 400 rpm for 2 h (4x30 min with 5 min stop in-between) in a ZrO₂ crucible (45 cm³) filled with 100 ZrO₂ balls (5 mm diameter).

How did you determine the cobalt content in the catalyst? How much Zn does the final Co_{0.5} catalyst contain? How much iron and zinc does the Fe_{0.5} catalyst contain?

From our past research on pyrolyzed Fe-N-C and Co-N-C catalysts, we know that Fe and Co do not form volatile products with such precursors in such pyrolysis conditions, and we have found that the Fe or Co content in catalysts (not subjected to any post-treatment) can be precisely estimated simply from the content before pyrolysis and the weight loss due to ZIF-8 and phen transformation into a N-doped host matrix. The absolute metal contents in pyrolyzed Fe-N-C or Co-N-C materials similarly prepared have been measured by various methods including neutron activation analysis (see Table 1 in Proietti *et al*, Nature Commun. 2 (2011) 416), ICP-MS (*Angew.* 54 (2015) 12753) and XAS (from the height of the absorption edge and catalyst loading of a pellet, *J. Electrochem. Soc.* 162 (2015) H403). The cobalt content in Co_{0.5} reported in the manuscript (1.5 wt %) is therefore an estimation from a previously proven method, involving the knowledge of Co content before pyrolysis and from the mass loss during pyrolysis (always measured for every synthesis).

Regarding the zinc content after pyrolysis, it is very small after pyrolysis at 1050°C in argon for 1 h since the boiling point of metallic Zn is only 907°C and also due to the fact that FeZn or CoZn alloys cannot be formed here since all Fe and Co atoms are integrated in isolated Metal-N_x-C_y moieties. In a first report on using ZIF-8 for preparing highly active Fe-N-C catalysts by pyrolysis, in which one of the corresponding authors took part (F. Jaouen), the Zn content was found to be only 0.06 at % after pyrolysis of ZIF-8, phenanthroline and Fe acetate at 1050°C in Ar for 1 h (see Table 1 in Proietti *et al*, Nature Commun. 2 (2011) 416).

Action: we only added "while Co does not form volatile compounds" in the Methods section on synthesis.

Was the weight loss of N-C similar to Co_{0.5} and Fe_{0.5}?

There is no significant difference in the weight loss due to ZIF-8 and phenanthroline transformation during pyrolysis in the absence of additional metal salt, and in the presence of cobalt acetate or of iron acetate at 0.5 wt % level. This was, for example, mentioned in our previous paper *J. Electrochem. Soc.* 162 (2015) H403. The mass of metal salt (Fe or Co acetate) corresponding to 0.5 wt% metal before pyrolysis is only *ca* 16 mg for 1000 mg of ZIF-8+phen (see Methods section) and therefore the overall mass loss observed during pyrolysis of catalyst precursors can be assigned almost entirely to ZIF-8 and phen.

Action: the following sentence in brackets has been added in the Methods section: "Due to a mass loss of 65-70 wt.% during pyrolysis in Ar (unmodified by the presence or absence of cobalt or iron at 0.5 wt % content) caused by volatile products formed from ZIF-8 and phen, the cobalt content in Co_{0.5} is 1.5 wt %"

Rotating (ring) disk electrode measurements:

- Is it correct that the ink of Pt/C is free of Nafion?

The ink for Pt/C also contained Nafion ionomer. In the first submitted version, we indicated by mistake our ink formulation for 45wt%Pt/C instead of for 5wt%Pt/C. We apologize for this. The ink formulation for 5%Pt/C was actually the same as used for Co_{0.5}.

Action: This is now described correctly in the methods section.

- Based on the description in ll. 361-362, Pt/C was directly measured in O₂ saturated electrolyte, without any activation step, is this right?

Pt catalysts must usually be electrochemically cleaned before ORR measurement. Here, the Pt/C electrode was cycled at 500 mV s⁻¹ between 0 and 1 V vs RHE for 300 cycles before performing the ORR measurement.

Action: This information has been added in the methods section.

- For background corrections always the initial scan in N₂ saturated electrolyte was used?

As a rule, and unless an unexpected signal or event occurred during the first scan in N₂-saturated electrolyte, we always used the second scan for the correction. The second scan usually superimposes with subsequent scans.

Action: The following sentence has been added to clarify this minor issue: "The second cycle was used for correction".

- The information on RRDE electrode is missing, what collection efficiency, what fabricate?

The RRDE electrode was from Pine Instruments (as the rotator). The collection efficiency indicated by the company for that Pt ring/glassy carbon disk RRDE is 0.37.

Action: This information has been added in the Supporting information.

Square-wave voltammetry:

- Information is missing on electrode area, ink volume and N/C ratio.

Action: The electrode area, volumes for water, isopropyl alcohol and Nafion solution in the ink are now indicated in the revised manuscript. The resulting Nafion (dry ionomer) to catalyst mass ratio is 10 wt % (also indicated in revised manuscript).

- Why did you select a lower catalyst loading compared to RRDE ?

A lower catalyst loading of $400 \mu\text{g cm}^{-2}$ was used for SWV compared to RRDE ($818 \mu\text{g cm}^{-2}$) because the intensity of the redox peak would be concealed by that of the double layer capacitance if we had used a higher catalyst loading. This is due to the high surface area of MOF-derived catalysts ($> 400 \text{ m}^2 \text{ g}^{-1}$). For the same reason, the redox peaks are absent in the classical CVs of these catalysts, and this is the reason why we resorted to the SWV technique to reduce the influence of the double layer capacitance (to increase the signal from redox peak to that of the capacitive current). Further optimization of the catalyst loading is however still needed to optimize the signal characteristic for the redox peaks by SWV.

Action: the following sentence was added: "This lower loading than used for most RRDE experiments was necessary to obtain a proper balance between the signal coming from the redox peak and that due to the double layer."

As SWV is correlated to in-situ XANES data in best case the ink should be similarly prepared and similar loadings should be used.

While the quantification of the electrocatalytic activity (and selectivity) toward ORR and toward other electrochemical reactions requires having a good control on the catalyst loading, the quantified information extracted from *in-situ* XANES (normalized spectra) does only little (or not at all, to some extent) depend on catalyst loading. We also do not expect the positions of the redox peaks (values of potentials) observed with SWV to depend much on the catalyst loading. Due to the narrow window of catalyst loading that is practically possible for SWV with such materials (as discussed in the reply to the previous question, above), we made our best to keep catalyst loadings used in RRDE, SWV and *operando* XANES in comparable ranges ($818, 400$ and *ca* $1000 \mu\text{g cm}^{-2}$, respectively).

Fuel cell:

Why were the flow rates of the gases not stoichiometric? Why 50 – 70 sccm (were the gas flows identical for all investigated catalysts?)

Typically, single -cell PEMFC tests performed in laboratory are designed so that the cell performance does not vary over the geometric area of the MEA (to characterize how the MEA works, and not be sensitive to the cell design itself). To do so, the flows of the anode and cathode gases are usually much higher than the amount of gas consumed at each electrode (even at the highest current density recorded during the polarization curve). The anode and cathode flows are therefore non stoichiometric with this approach (not in a 2:1 ratio), but need to be highly overstoichiometric (incoming flow much higher than the gas electrochemically consumed in each electrode when the PEMFC operates). The flux at the cell exit was controlled at 50 sccm (same value for anode and cathode exit). This allows to have identical gas velocity in the gas channels of the PEMFC, and therefore similar driving force to remove any water droplets (important, as the incoming gases are saturated in water vapor). The range 50-70 sccm indicated in the first submission corresponds to the flux at the cell inlet. As the exiting fluxes are regulated, the flux at inlet progressively increase above 50 sccm to compensate for the consumption rate of gases (i.e. proportional to the current density at any time), and therefore we indicated 50-70 sccm, with 70 sccm typically corresponding to the inlet flux observed at maximum current density.

Action: we agree that the range 50-70 sccm indicated was confusing and we modified the text to: "The flow rates for humidified H₂ and O₂ were controlled downstream of the PEMFC and fixed at 50 sccm".

Operando XAS

□ Line 392: I guess you mean Fe K-edge and Co K-edge...

Yes, both Fe K-edge and Co K-edge spectra were recorded at beamline SAMBA and with the same equipment. The text has been modified on the former line 392 as suggested by the reviewer.

□ What active area did the graphite electrode had? The ink preparation, catalyst loading etc. is not precise

The circular area of the graphite electrode in contact with the electrolyte was 4.9 cm², but it was only partially covered by the catalyst ink (see photos below as an example). A central area of *ca* 3 cm² was covered by the 50 μL aliquot of catalyst ink, resulting in a local catalyst loading of *ca* 1 mg cm⁻². A significantly thicker loading would increase the risk of delamination of the catalyst layer from the graphite foil (formation of "dendrites") and the risk of uneven deposition. More advanced deposition methods to fully cover the graphite foil for *operando* XANES would be welcome, but is difficult to have access to at synchrotron facilities while doing the measurements. Besides, the only practical advantage to have the whole graphite electrode catalyzed would reside in the lack of need to position the X-ray beam before acquiring spectra, saving only a small amount of time.

Action: the methods section now describes the loading used for operando XANES measurement and the cell, and an additional supporting figure S14 shows the cell design and setup used.

What detector was used for fluorescence measurements?

A Canberra 35-elements monolithic planar Ge pixel array detector.

Action: this information has been added in the Methods section.

Please provide more detailed information on step size etc. for XAS

Co K-edge spectra have been measured between 7560 and 8400 eV with an energy step of 5 eV in the pre-edge region up to 7640 eV, then 1 eV up to 7680 eV, 0.2 eV up to 7750 eV, 0.5 eV up to 7850 eV, and 2 eV over the rest of the spectrum.

The integration time used is 1 second between 7560 and 7680 eV, and 2 seconds in the rest of the acquisition.

Fe K-edge spectra have been measured between 6950 and 7800 eV with an energy step of 5 eV in the pre-edge region up to 7050 eV, then 1 eV up to 7080 eV, 0.2 eV up to 7150 eV, 0.5 eV up to 7250 eV, and 2 eV over the rest of the spectrum. The integration time used is 1 second between 6950 and 7080 eV, and 2 seconds in the rest of the acquisition.

Action: while the above mentioned information might be useful, it risks overloading the text; therefore it has not been included in the manuscript.

□ Line 441: is there any logic behind the value $\varepsilon = 1.2\%$?

The experimental error ε (assumed to be constant and 1.2% of the main experimental edge jump over the whole data set) has been estimated assuming that the data is dominated by noise that is independent of R in the R -space: the white noise. This method was proposed by Newville et al. [J. Synchrotron Rad. 6, 264-265 (1999)]. The starting point is the assumption that the FT of $\chi(k)$ at long distance does not have any structural signal and is only due to the noise ε . The algorithm is implemented in data analysis tools like Viper [K. V. Klementiev, VIPER for Windows, freeware; K. V. Klementiev, J. Phys. D: Appl. Phys. 34, 209-17 (2001)] used to estimate the ε in this work.

SQUID:

Where does the magnetic moments in N-C come from?

Graphite itself is a known diamagnetic material (negative magnetic susceptibility) with anisotropic behavior (larger diamagnetism perpendicular to the plane) (e.g. Phys. Rev. B 49 (1994) 15122). The magnetic properties of graphite and of other carbon allotropes are strongly related to whether or not aromatic π electrons are present. The nitrogen doped carbon N-C is a complex material comprising amorphous carbon (not only six-membered rings of carbon atoms) with various nitrogen groups but also typically some oxygen functional groups (after air exposure) that may contribute to the overall magnetic behavior observed. The overall magnetic susceptibility of N-C is in fact slightly negative at $T > 100$ K (Fig. S7 in the first submitted manuscript). Paramagnetism and even ferromagnetism of doped graphite (nitrogen doped graphite in particular) has however been reported in the literature (see e.g. Q. Miao *et al.*, Scientific Reports 6 (2016), Art. Nbr. 21832) and is assigned to defects. This research topic is however controversial, with very small amount of metallic impurities possibly being responsible for such phenomena. The very small amount of zinc present after pyrolysis of ZIF-8 in our work may for example explain the magnetic properties for N-C reported in Fig. S7 of the first submitted version. The presence of minor content of Fe impurities in commercial ZIF-8 under tradename Basolite Z1200 (ca 50-100 ppm by weight, measured by us (not published) or others, see e.g. Nano Energy 29 (2016) 111) also impedes the measurement of magnetic properties of a pure N-C material prepared from this ZIF-8 here. The level of Fe impurities is however sufficiently small to neglect its impact on the overall electrochemical activity of Co-N-C (0.5wt% Co before pyrolysis corresponds to 5000 ppm Co by weight).

For all those reasons, we prefer not discussing the origin of magnetic properties of N-C in the main manuscript, and we simply extracted the magnetic signal due to isolated cobalt atoms by subtracting the signal measured for Co-N-C by that of N-C. The Co-N-C material can be considered to magnetically behave as the sum of N-C and of cobalt moieties because:

- i) The graphitic degree and material structure are similar (XRD, XPS are similar). This similarity is due to the atomic dispersion of cobalt, while any agglomeration of cobalt into nanoparticles would catalyze graphitization during the pyrolysis process.
- ii) The content of Zn impurities in N-C and Co-N-C are similar due to i) above and due to the identical pyrolysis temperature and duration used to prepare them, and due to the fact that no acid leaching was applied after pyrolysis

DFT

Calculations were made for cobalt, only?

The same calculations for various FeN_xC_y moieties have been performed by us previously (Zitolo et al. Nature Mater. 14 (2015) 937). The figure S6 in the first submitted version shows the four model clusters on which DFT was applied for cobalt moieties: $\text{CoN}_4\text{C}_{12}$, $\text{CoN}_4\text{C}_{10}$, $\text{CoN}_3\text{C}_{10,\text{porp}}$ and CoN_2C_5 . Except for $\text{CoN}_4\text{C}_{10}$ (included because it is a cobalt moiety often believed to be present in pyrolyzed materials), these cobalt moieties were selected for DFT investigation because they can reproduce the experimental XANES spectra of the model catalyst $\text{Co}_{0.5}$. The similar reasoning was adopted in Nature Mater. 14 (2015) 937 to select Fe-based moieties for DFT investigation: $\text{FeN}_4\text{C}_{12}$ was included because it can reproduce the experimental XANES spectra of a model Fe-N-C catalyst free of Fe particles.

While the moiety $\text{CoN}_3\text{C}_{10,\text{porp}}$ can reproduce the XANES experimental spectrum of the model cobalt catalyst, we verified in the present work that $\text{FeN}_3\text{C}_{10,\text{porp}}$ and FeN_2C_5 cannot reproduce the XANES experimental spectrum of the model Fe-N-C catalyst (Figure S8 in the first submitted version of the manuscript), and therefore we do not feel that it is interesting to include in the manuscript the DFT results for $\text{FeN}_3\text{C}_{10,\text{porp}}$ and FeN_2C_5 .

Reviewers' Comments:

Reviewer #1:

Remarks to the Author:

The authors have properly addressed all of my questions and modified the text accordingly.

Therefore, I gladly recommend their article now for publication in Nature Communications.

I just have one minor point: to a comment of mine that started "Moreover, if the authors intend to extrapolate from the results of molecular catalysts..." the authors provided a fine reply but did not take any action about it in the text. For the sake of discussion, I kindly suggest that they include it together with suitable citations.

Reviewer #2:

Remarks to the Author:

The authors have taken a lot of effort to understand the comments of reviewers and to use them to improve the manuscript. Overall, the new version has a more clear structure, more complete experimental details, and a better connection between the presented results and conclusions.

The presentation of OER activity, however, could still be improved. The claim of 1.25 V onset potential for OER remains to be misleading. Because the thermodynamic potential for the reaction is at 1.23 V, defining the onset potential at 1.25 V does little to differentiate between various catalysts. Furthermore, the activity remains low until about 1.6 V, at which point it starts to increase exponentially. For comparison, when looking at HER activity, the authors do not claim that the onset potential is close to 0 V, but rather state that the activity is shifted by 200 mV relative to Pt. A similar comparison could be made for OER, but with using IrOx benchmark. Finally, it is important to consider that no proof is presented that the current below 1.6 V is associated with production of O₂. A possible alternative is that carbon corrosion current increases in the presence of Co catalyst.

Reviewer #3:

Remarks to the Author:

So far I understood the novel, important and groundbreaking conclusion that should make it worth being publishable in this journal is the fact that the O₂ adsorption strength on cobalt-based moieties is too weakly for efficient O₂ reduction?

If so, the manuscript is ready for publication.

REVIEWERS' COMMENTS:

Reviewer #1 (Remarks to the Author):

The authors have properly addressed all of my questions and modified the text accordingly. Therefore, I gladly recommend their article now for publication in Nature Communications.

I just have one minor point: to a comment of mine that started "Moreover, if the authors intend to extrapolate from the results of molecular catalysts..." the authors provided a fine reply but did not take any action about it in the text. For the sake of discussion, I kindly suggest that they include it together with suitable citations.

Answer: The main reason why we did not include additional text in the first revision of the main manuscript on this interesting question and discussion is space limitation. We have now added some text and cite the work reported in Surface Science (new reference 50 in the second revision).

Action: the following text has been added on page 14:

"Last, while general reactivity trends seem to apply for both pyrolyzed Metal-N-C materials and well-defined metal macrocycles, this does not mean however that their absolute ORR activity and O₂ adsorption energy are the same. For example, the O₂ adsorption energy on the three porphyrinic-derived CoN_xC_y moieties in ground state is -0.80 to -1.26 eV (**Supplementary Table 4**), higher than that reported for cobalt phthalocyanines and porphyrins (-0.60 to -0.35 eV).⁴² The covalent integration of the Metal-N_x moieties in a conductive carbon matrix generally modifies the electron density at the metal relative to a metal macrocycle adsorbed on carbon.⁵⁰"

Reviewer #2 (Remarks to the Author):

The authors have taken a lot of effort to understand the comments of reviewers and to use them to improve the manuscript. Overall, the new version has a more clear structure, more complete experimental details, and a better connection between the presented results and conclusions.

The presentation of OER activity, however, could still be improved. The claim of 1.25 V onset potential for OER remains to be misleading. Because the thermodynamic potential for the reaction is at 1.23 V, defining the onset potential at 1.25 V does little to differentiate between various catalysts.

Regarding the onset of OER on Co_{0.5}, we agree with the reviewer that it is not clearly defined, and we have modified the sentence to be more rigorous: "The onset of OER in acidic medium on Co_{0.5} is not well defined but clearly situated above the redox peak associated with Co(II)/Co(III) at 1.25 V vs. RHE (**Supplementary Figure 2**), suggesting Co(III)N_xC_y moieties could be the active site for OER in Co_{0.5}"

Furthermore, the activity remains low until about 1.6 V, at which point it starts to increase exponentially. For comparison, when looking at HER activity, the authors do not claim that the onset potential is close to 0 V, but rather state that the activity is shifted by 200 mV relative to Pt. A similar comparison could be made for OER, but with using IrOx benchmark.

Answer: we have added a recently measured reference curve for OER with IrO₂ nanoparticles made in our laboratory (CNRS) in supplementary figure 10. We have also modified the main text to report OER activity following the suggestion of the reviewer.

New sentence: "Due to the low density of cobalt sites in Co_{0.5}, the polarization curve is however positively shifted by 200 mV relative to that of unsupported IrO₂ nanoparticles at a current density of 2 mA cm⁻²."

Finally, it is important to consider that no proof is presented that the current below 1.6 V is associated with production of O₂. A possible alternative is that carbon corrosion current increases in the presence of Co catalyst.

Regarding the idea that cobalt in Co-N-C could catalyze the carbon corrosion and thus could lead to faster corrosion than on N-C, it is a possibility that we cannot exclude at this moment (it could also occur, but not necessarily accounting for 100% of the current measured at low potential). A sentence has been added to highlight that the current in the 1.2-1.6 V vs RHE range measured on Co_{0.5} could be due partly or entirely to carbon corrosion. We have added the sentence: "The low current observed at 1.3-1.6 V vs. RHE on Co_{0.5} might in fact entirely or partially originate from carbon corrosion, enhanced in the presence of cobalt relative to N-C."

Reviewer #3 (Remarks to the Author):

So far I understood the novel, important and groundbreaking conclusion that should make it worth being publishable in this journal is the fact that the O₂ adsorption strength on cobalt-based moieties is too weakly for efficient O₂ reduction?

If so, the manuscript is ready for publication.

Answer: In our opinion, the first important result is the demonstration here that the cobalt sites in pyrolyzed materials have a very similar structure to the iron sites in pyrolyzed materials, and also similar structure to metal porphyrins. This has not been demonstrated before for cobalt-N-C. The second important result is the demonstration (from theory and from experimental voltammetry and operando XANES spectroscopy results) that cobalt sites bind O₂ more weakly than iron sites, in line with the trends previously observed for iron and cobalt macrocycles adsorbed on carbon. The combination of both results brings a comprehensive understanding of the site structure and reactivity toward ORR of Fe-N-C and Co-N-C pyrolyzed materials, improving the knowledge on such materials one step ahead. It also clearly opens the path for the site-structure determination and volcano plot determination for an even broader set of metal-N-C materials in the near future.